# Robustness to 3D Object Transformations in Humans and Image-Based Deep Neural Networks

**Haider Al-Tahan**
Neuroscience Graduate Program
Department of Psychology
Western University
London, ON, Canada
haltaha@uwo.ca

**Farzad Shayanfar**
Research Volunteer
Western University
London, ON, Canada
farzad.shayanfar@hotmail.com

**Ehsan Tousi**
Neuroscience Graduate Program
Department of Psychology
Western University
London, ON, Canada
ekahooka@uwo.ca

**Marieke Mur**
Department of Psychology
Department of Computer Science
Western University
London, ON, Canada
mmur@uwo.ca

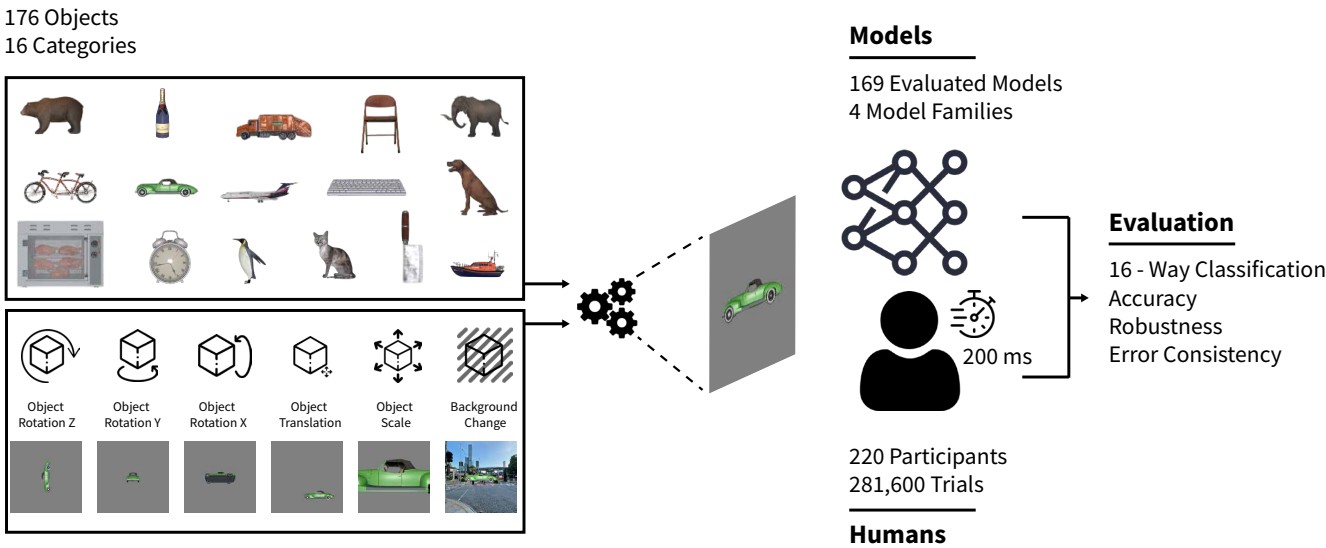

Figure 1: **How robust are computational models to 3D object transformations compared to humans?** We evaluate 169 models across six identity-preserving 3D transformations: object rotation, scaling, translation, and background change on 176 3D objects. We benchmark these models against humans, analyzing 281,600 trials collected from over 220 participants.

## Abstract

**Recent work at the intersection of psychology, neuroscience, and computer vision has advocated for the use of more realistic visual tasks in modeling human vision. Deep neural networks have become leading models of the primate visual system. However, their behavior under identity-preserving 3D object transformations, such as translation, scaling, and rotation, has not been thoroughly compared to humans. Here, we evaluate both humans and image-based deep neural networks, including vision-only and vision-language models trained with supervised, self-supervised, or weakly supervised ob-jectives, on their ability to recognize objects undergoing such transformations. Humans (n=220 ) and models (n=169 ) were asked to categorize images of 3D objects, generated with a custom pipeline, into 16 object categories recognizable by both. Humans were time-limited to reduce reliance on recurrent processing. We find that both humans and models are robust to translation and scaling, but models struggle more with object rotation and are more sensitive to contextual changes. Humans and models agree on which in-depth object rotations are most challenging – when humans struggle, models do too – but humans are more robust and show**

Table 1: **Comparison with existing benchmarks.** Unlike prior work, our dataset provides human data and ensures controlled variation across transformations while maintaining a large number of objects.

| Benchmark | Human Data | Controlled Variation | Number of Transformations | Number of 3D Objects | Task |
|---|---|---|---|---|---|
| ObjectNet (Barbu et al., 2019) | ✗ | ✗ | 3 | ∼ | Categorization |
| NVD (Ruiz et al., 2022) | ✗ | ✗ | 5 | 92 | Categorization |
| PUG: ImageNet (Bordes et al., 2023) | ✗ | ✗ | 10 | 724 | Categorization |
| MOCHI(Bonnen et al., 2024) | 35k trials | ✗ | 3 | ∼ | Identification |
| ORBIT | 281k trials | ✓ | 6 | 176 | Categorization |

**more consistent category confusions with one another than with any model. By testing model families trained on different amounts of data and with different learning objectives, we show that data richness plays a substantial role in supporting robustness – potentially more so than vision-language alignment. Our benchmark excludes models trained on video, multiview, or 3D data, but is in principle compatible with such models and may support their evaluation in future work. This study underscores the importance of using naturalistic visual tasks to model human object perception in complex, real-world scenarios, and introduces a benchmark - ORBIT (Object Recognition Benchmark for Invariance to Transformations) - for evaluating and developing computational models of human object recognition. Code and data for ORBIT are available at: `https://github.com/haideraltahan/ORBIT`.**

**Keywords:** invariant object recognition; 3D object transformations; robustness; model-to-human alignment; deep neural networks; human behavior

## Introduction

To develop accurate models of the primate visual system, it is essential to evaluate models on naturalistic visual tasks that reflect the complexity of object perception (Peters & Kriegeskorte, 2021; Bowers et al., 2022). One such task is image categorization across 3D object transformations, which is central to primate vision (Marr, 2010; DiCarlo & Cox, 2007) and presents a more naturalistic challenge than standard benchmarks (Deng et al., 2009). Recognizing objects across transformations like rotation, scaling, and translation is fundamental to human visual processing, yet remains challenging for artificial models to replicate. By focusing on these transformations, we bring models closer to real-world visual challenges.

Recent work has begun to systematically compare human and neural network performance on view-invariant object recognition, revealing important differences in how each system handles this task (O'Connell et al., 2023; Bonnen et al., 2024; Ollikka et al., 2025). Even when models achieve high accuracy, their behavior may not match human perceptual strategies (Geirhos et al., 2020). Often, models rely on simpler or alternative processing shortcuts that diverge from the

more complex, dynamic strategies humans use to recognize objects across different views (Bonnen et al., 2024). These findings highlight the need for closer alignment between how models and humans process visual information.

To better understand this alignment, we introduce **ORBIT** (Object Recognition Benchmark for Invariance to Transformations), a new benchmark that evaluates both models and humans on object categorization under 3D transformations (Figure 1). Unlike previous datasets (Table 1), ORBIT provides controlled variation across six transformations, a large set of 3D objects, and 281k human response trials from 220 individuals, enabling rigorous comparison of model robustness and human perception. We focus on deep neural networks that take a single static image as input and support object categorization, including vision-only and vision-language models trained with (weakly) supervised or self-supervised objectives. By systematically varying transformation parameters, we investigate where models succeed, where they fail, and how their errors compare to humans, offering new insights into the alignment between artificial and biological vision.

## Methods

### Task

Humans and models performed the same task: a forced-choice categorization task on object images. While DNNs, typically trained on the ImageNet dataset (Deng et al., 2009), classify objects into a large set of fine-grained categories, human perception tends to favor broader, more intuitive "basic-level" categories (Rosch et al., 1976; Geirhos, Janssen, et al., 2018). To align human and model label spaces, we employed a mapping approach proposed by (Geirhos, Temme, et al., 2018), where the 1,000 fine-grained ImageNet categories are consolidated into 16 broader categories using the WordNet hierarchy (Fellbaum, 2010). These categories include airplane, bear, bicycle, bird, boat, bottle, car, cat, chair, clock, dog, elephant, keyboard, knife, oven, and truck.

### Stimuli

**3D object stimuli.** The stimuli used in ORBIT are rendered images of 3D objects shown under a range of identity-preserving transformations. The images were generated using 176 3D objects sourced from prominent online repositories

(Tables 6 to 9). Object selection was guided by two key criteria: relevance to the 16 predefined categories and high visual realism. Only models that clearly fit the broader category labels were included to ensure consistency across the dataset. Each object was required to be high-resolution and visually detailed (e.g., with a high number of polygons). To standardize the stimulus set for consistent evaluation, we applied two preprocessing steps to each 3D object. First, we performed uniform scaling by resizing models based on their largest bounding-box dimension, ensuring consistent size while preserving proportions. Second, we applied object centering by translating each model to the center of the camera's coordinate frame, standardizing position across renderings. Figure 6 provides a mosaic of all objects in their canonical views.

**Identity-preserving transformations.** To generate the full image set, we used Unity, a 3D game engine, to render each object under a controlled set of transformations designed to probe robustness in object recognition. To facilitate stimulus generation, we developed `augment3D`, an open-source package to create image datasets for human and model experiments on object vision, which we plan to release later this year. The transformations we applied included: object translation, scaling, rotation, and background changes. Translation was implemented by repositioning objects on a polar grid, varying direction and distance from the image center. Scale was varied by adjusting the camera distance, simulating zoom effects from 12.5% to 200% of baseline object size. Rotation was applied around each world axis in 45° steps, keeping the object centered and camera fixed. Backgrounds were changed using six natural scene images categorized as congruent or incongruent with each object class, alongside a colored noise control and a neutral gray baseline matched to ImageNet's average pixel values. To illustrate the range of applied transformations, Figure 7 shows an example object rendered under each transformation.

**Evaluation images.** Each transformation was applied independently to all objects, resulting in a balanced and fully crossed stimulus set. A single canonical view was used as the baseline across all transformations, accompanied by seven transformed views per object for each transformation condition, yielding 1,280 images per transformation (16 categories × 10 objects × 8 images). Across all transformations, this resulted in a total of 7,680 images, each rendered at a resolution of 224 × 224 pixels. Further implementation details are provided in the *Appendix*. This systematic and diverse synthetic image dataset forms a comprehensive benchmark for evaluating both human and model performance on invariant object recognition.

### Human experiment

**Participants.** A total of 226 participants completed the experiment. Participants were recruited via Prolific, an online participant recruitment platform. Eligible individuals were between 18 and 35 years of age, had normal or corrected-to-normal vision, and reported no history of neurological disorders. Prior to participating in the study, all individuals provided implied consent through a questionnaire administered on Qualtrics (Qualtrics, Provo, UT). Participants were compensated at a rate of $15 per hour. All procedures were approved by the Office for Research and Ethics at Western University.

To ensure data quality, we included an ImageNet test block prior to the main experimental blocks. Participants who did not reach at least 80% accuracy on this test were not invited to continue and are not included in the final sample. Additionally, six participants were excluded from the analysis because their performance fell more than 1.5 times the interquartile range below the median.

**Experimental design.** Each trial began with a gray screen displaying a central white fixation cross. Participants initiated the trial by clicking on the fixation cross. Following this action, a stimulus image was presented for 200 ms, immediately followed by a colored mask lasting 200 ms. The mask served to increase task difficulty and reduce the influence of recurrent processing on performance. Next, a response screen appeared, displaying multiple category buttons arranged in a circular configuration around the fixation point. This screen remained visible for up to 1500 ms or until the participant responded, whichever occurred first. The circular layout ensured equal distance from the fixation cross to each button, minimizing potential spatial biases in decision making. To avoid anticipatory movements, the mouse cursor was locked during image and mask presentation. During the response phase, participants selected the category they deemed most congruent with the stimulus (Figure 8). More details on the human experiment can be found in the *Appendix*.

### Neural network models

We conducted a systematic analysis of 169 models and distilled this set to a representative subset of 31 models for focused comparison with human behavior. This subset was selected to span a broad range of architectures, training objectives, and dataset scales, while minimizing redundancy in the main figures. We selected models that (1) have been widely used in cognitive computational neuroscience, (2) represent distinct trends in modern model development (e.g., transformer-based vision models, vision-language pretraining), and (3) span meaningful variation in architecture, learning objective, and data scale. While these dimensions are not fully disentangled – model families, as defined below, represent generational shifts that co-vary along multiple axes – they nonetheless offer insight into which aspects of model design may contribute to human-like robustness. All model evaluations, including those for the full set of 169 models, are available in the supplementary codebase: `https://github.com/haideraltahan/ORBIT`. The 31 selected models fall into four families:

1. **Vision-only**: This category comprises early convolutional neural networks (CNNs), including AlexNet (Krizhevsky et al., 2012), VGG-16 (Simonyan & Zisserman, 2015), ResNet50 (He et al., 2015), and SqueezeNet (Iandola et al., 2016), along with early vision transformers (ViT-B/32)

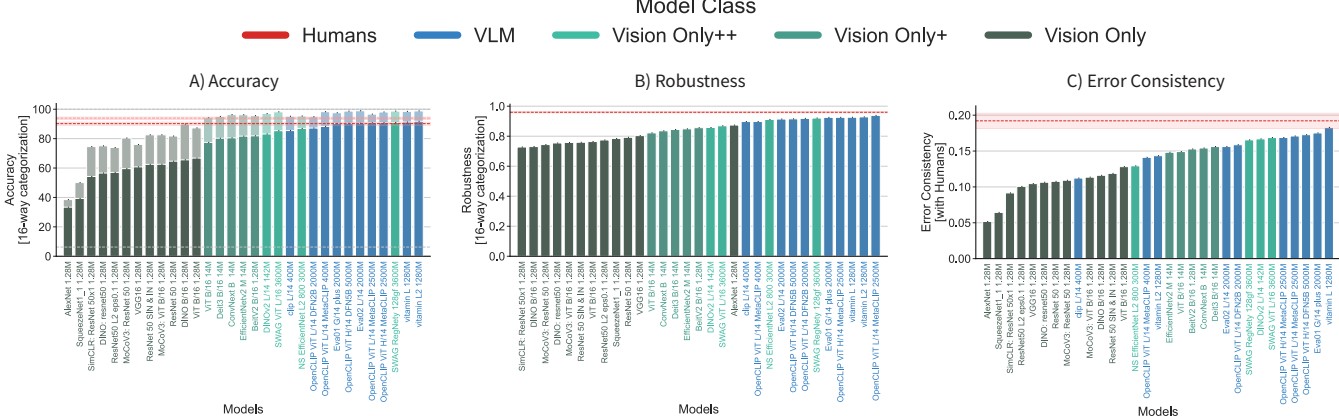

Figure 2: **Human and model performance averaged across transformations.** Human performance is shown using red dotted lines, with shaded areas indicating 95% confidence intervals. (A) Accuracy at object categorization for canonical views (light bars) and transformed views (dark bars). (B) Robustness, computed by expressing accuracy for transformed views as a proportion of accuracy for canonical views. (C) Error consistency. Bars show the average consistency between each model and individual human participants. The red dotted line indicates average consistency between individual human participants.

(Dosovitskiy et al., 2021). These models were trained on ImageNet-1K using either categorical supervision or self-supervised learning objectives such as contrastive learning. Self-supervised models within this category underwent linear probing on ImageNet-1K. Additionally, we included implementations of ResNet50 designed to enhance robustness, although they do not explicitly account for transformations in 3D object space (Salman et al., 2020).

2. **Vision-only+**: This category comprises more advanced vision architectures, including ConvNext (Liu, Mao, et al., 2022), EfficientNet (Tan & Le, 2021, 2019), and DeiT III (Touvron et al., 2022), trained on the ImageNet-21K (Ridnik et al., 2021) dataset. These models employ categorical supervision or self-supervised methodologies, frequently incorporating knowledge distillation (Touvron et al., 2022) and masked image modeling with semantic targets (Peng et al., 2022). Self-supervised models in this category were further fine-tuned on ImageNet-1K. Both vision-only and vision-only+ models are widely utilized in computational neuroscience due to their alignment with hierarchical visual processing in biological systems (Conwell et al., 2024).

3. **Vision-only++**: This category encompasses large-scale vision models trained on datasets exceeding 100 million images through weakly supervised learning paradigms. These models leverage noisy supervisory signals, such as human-generated hashtags Singh et al. (2022) or teacher-generated labels (Q. Xie et al., 2020), to optimize feature representations at scale. These models offer insight into the role of large-scale data exposure in supporting robustness to 3D transformations

4. **Vision-Language Models (VLMs)**: This category includes multimodal architectures trained using contrastive

language-image pretraining (CLIP) (Radford et al., 2021), which enables them to learn rich, cross-modal representations that bridge visual and linguistic information. Recent studies suggest that VLMs exhibit strong alignment with neural activity in high-level visual cortex (Wang et al., 2023), a region crucial for forming abstract, invariant representations of objects across diverse transformations (DiCarlo & Cox, 2007; Rust & DiCarlo, 2010). Given their exposure to large-scale image-text data, VLMs may develop more flexible object representations, supporting robustness to identity-preserving transformations. However, whether their improved performance is driven by semantic enrichment from language supervision or simply greater dataset scale remains an open question, motivating comparisons with vision-only models trained on similarly large datasets.

### Evaluation metrics

To assess performance across humans and DNNs, we employed three evaluation metrics:

1. **Accuracy:** Proportion of correctly categorized stimuli. For human participants, accuracy corresponds to the proportion of trials in which the selected category matched the ground truth (one of the 16 predefined categories). For models trained or linearly probed on ImageNet, we mapped their 1,000-class predictions to the 16 target categories (Geirhos, Temme, et al., 2018). Similarly, for VLMs, we computed zero-shot accuracy by first acquiring predictions on the 1,000 ImageNet classes, then mapping to the 16-category scheme.

2. **Robustness:** Measure of the ability to maintain performance across transformations, reflecting generalization capacity under changes in object appearance. Robustness is computed as the ratio of accuracy on transformed views

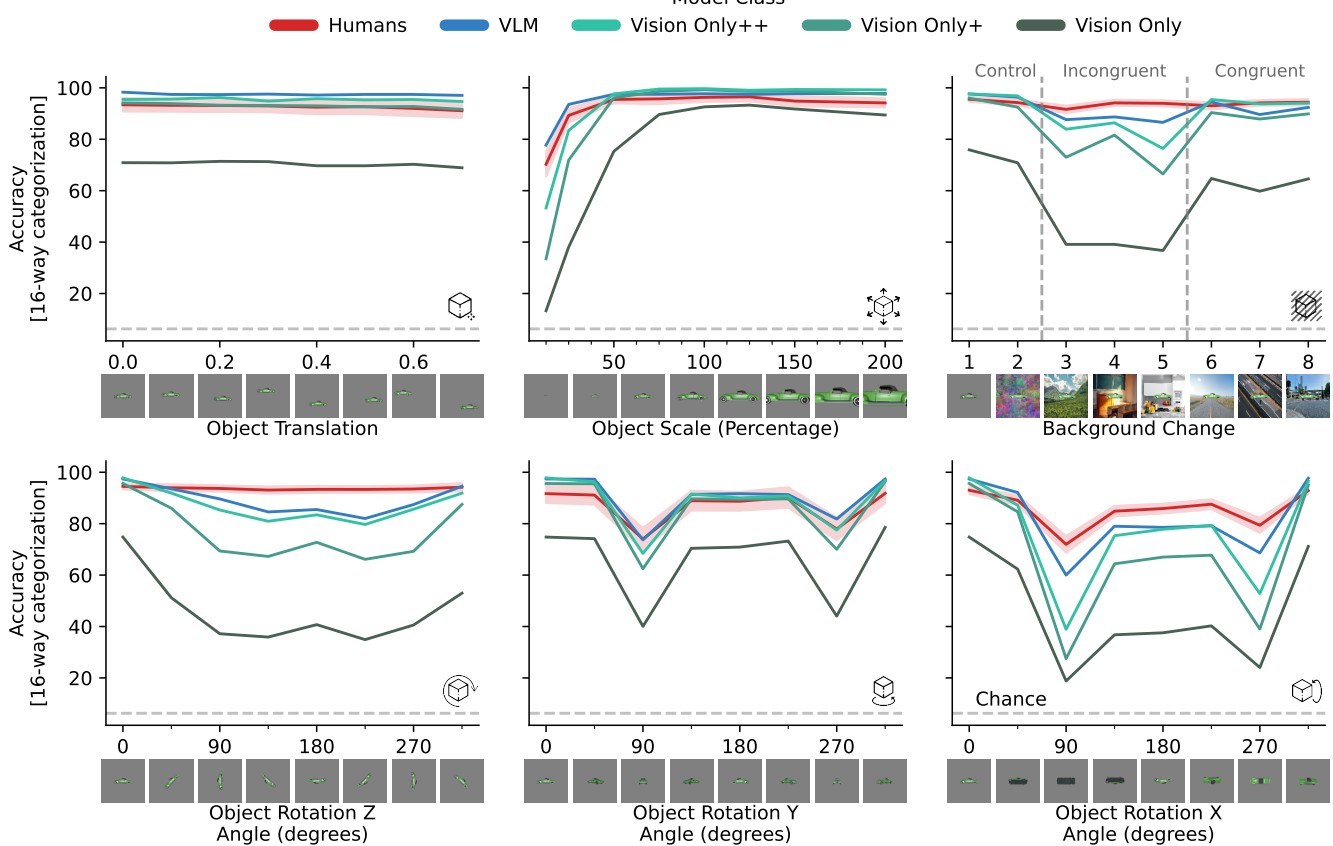

**Figure 3: Human and model performance as a function of transformation level across different transformations.** Each panel shows accuracy at 16-way categorization for a specific transformation: object translation, object scaling, background change, in-plane object rotation (z), in-depth object rotation (y), and in-depth object rotation (x). The dashed gray line represents chance performance. The colored lines show mean performance across humans (red, with 95% confidence intervals) or models in a family (blues and greens).

to accuracy on canonical views of the same objects. A robustness score close to 1 indicates strong invariance, while lower/higher values indicate sensitivity to transformations. This metric captures how well models and human participants maintain recognition performance across changes in viewpoint, scale, position, and background context.

3. **Error consistency:** The similarity of misclassification patterns between two decision makers (e.g., different models or human participants). High overall accuracy can lead to high observed consistency even if the errors occur on different images. To account for this, we used a chance-adjusted measure (Geirhos et al., 2021, 2020) to evaluate whether the observed consistency exceeds what would be expected from independent decision makers with matched accuracy.

## Results

### Data-rich models are closing the gap with time-limited humans

We computed categorization performance for humans and models for canonical and transformed views of the objects. Canonical views were a priori determined by the experimenters and shown in Figure 6 for an example object. Figure 2A shows performance averaged across all identity-preserving transformations. While vision-only models are below human performance on canonical as well as transformed views, vision-only+ models reach human-level performance on canonical views, and some vision-only++ models and most VLMs reach human-level performance on transformed views. However, the relative drop in performance when moving from canonical to transformed views appears smaller for humans than models, i.e. humans appear more robust. We next computed robustness for humans and models by expressing performance for transformed views as a ratio of the performance for canonical views. Figure 2B confirms our observation that

models are approaching human-level robustness, with VLMs leading. VLMs are also leading on error consistency with humans (Figure 2C), suggesting that these models may use similar strategies as humans when categorizing objects across identity-preserving transformations.

VLMs are the top-ranked models on all measures shown in Figure 2. However, it is important to note that the included vision-language models are data-rich models trained on millions or billions of images. To assess the impact of visual diet, we evaluated a subset of vision-language models with identical architectures trained with CLIP using varying amounts of data. The amount of training data has a considerable impact on accuracy (Figure 4). Accuracy monotonically increases with the amount of training data, in a similar fashion as observed for the vision-only models in Figure 2.

Beyond dataset size, model complexity also plays a significant role in performance. As shown in Figure 4, increasing model size – measured by the number of parameters – improves categorization accuracy, with larger architectures (e.g., Eva-01 G/14 1012M) approaching human-level performance. This pattern is observed across both vision-language models and vision-only++ models, suggesting that both larger-scale training and more complex architectures contribute to improvements in object recognition under transformations.

Consistent with these observations, vision-only++ models, which are trained on similarly large amounts of images as the VLMs, show accuracy and robustness comparable to their vision-language-trained counterparts (Figure 2AB). These results suggest that data richness is a stronger determinant of robustness to identity-preserving object transformations than vision-language alignment, and model complexity further enhances performance.

While the gap is closing, models do not fully achieve human-level robustness to identity-preserving transformations. We next investigated which transformations are contributing to this gap. Accuracy, robustness, and error consistency metrics for individual transformations suggest that object rotation and changes in background are driving the robustness gap (Figures 9 to 11).

## Object rotation is challenging for both humans and models, but humans are more robust

To get a more detailed picture of model-to-human alignment, we plotted performance at object categorization as a function of transformation level, for each transformation separately (Figure 3). These plots confirm that humans and models are robust to object translation and, to a large extent, object scaling, but are challenged by object rotation.

For in-plane object rotation, models exhibit a decline in accuracy as rotation increases, with performance dipping most strongly around 180 degrees away from the canonical orientation. In contrast, humans maintain high accuracy across all rotation angles, suggesting that they are robust to this transformation. The data-rich vision-only++ models and VLMs get closest to humans, while the earlier generations of vision-only

models do noticeably worse. Despite the differences in robustness across model families, they do agree on which rotations are most challenging.

In-depth object rotation is challenging for both humans and models (Figure 3). Their accuracy decreases notably at 90 and 270 degrees away from the canonical orientation, where an object is seen from the front and back, or bottom and top. Interestingly, the data-rich vision-only++ models and VLMs reach human-level performance when an object is rotated about its vertical axis (y; front-back) but not its horizontal axis (x; bottom-top). Performance differences between model families are also larger for rotation about the horizontal axis. These observations suggest that models struggle more with occlusion and perspective changes introduced by this transformation. We speculate that the models' visual diet may contain more images of objects rotated about their vertical than their horizontal axis. In sum, humans and models are challenged by object rotation in-depth and agree on which rotations are most difficult, but humans are more robust.

While these results highlight the overall robustness advantage of humans, closer inspection reveals that the performance gap may narrow under certain conditions. Figure 12 shows that for in-depth rotation about the horizontal axis (x), the top-performing models closely approach human performance, while a consistent gap remains for in-plane rotation. To better understand the source of the remaining performance gap, we analyzed human accuracy over the course of the experiment. Accuracy improved across blocks, particularly for these two challenging transformations. Notably, during the first block of trials, human accuracy was comparable to that of the best-performing models for both in-plane and in-depth rotation, suggesting that part of the human advantage may reflect memory or learning effects accumulating over time (see Figure 13).

## Humans are less context dependent than models

Humans demonstrate greater robustness to background changes than models, which exhibit a noticeable drop in accuracy when objects appear in incongruent backgrounds (e.g., a car placed in an indoor setting) (Figure 3). This suggests that models – particularly vision-only models – rely on background context for object categorization, likely due to strong statistical correlations in their training data between objects and their typical environments.

This context dependence is most pronounced in vision-only models, which show the largest performance drop when background variations are introduced. As seen in Figure 5, models exhibit greater variability across object categories than humans, with certain categories being disproportionately affected by background changes. This suggests that these models may incorporate spurious background cues into their predictions rather than relying purely on object-intrinsic features, making them vulnerable to shifts in contextual information.

In contrast, data-rich models, including vision-only++ models and VLMs, exhibit improved robustness to background changes. Several top-performing models even approach

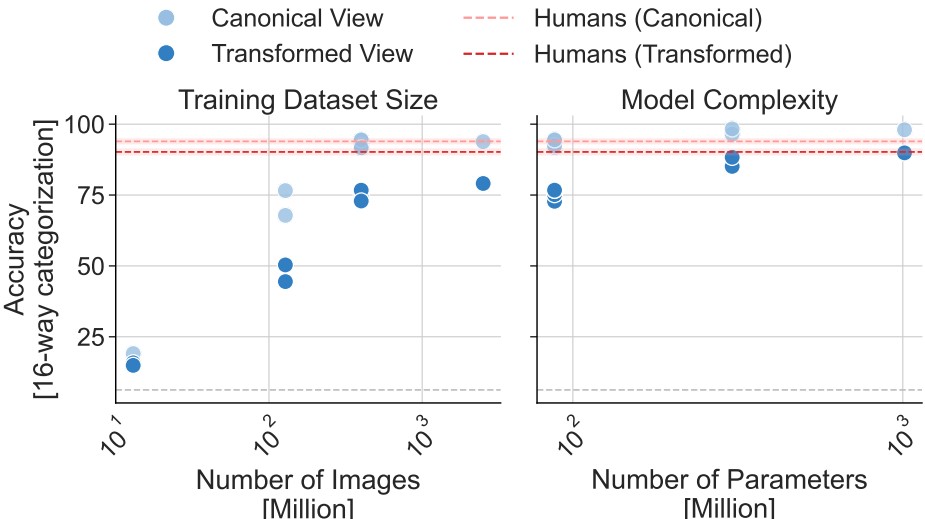

Figure 4: **Accuracy as a function of training dataset size and model complexity.** Dots represent different CLIP models, varying in training dataset size (left) and model complexity (right). Each model contributes two dots: light blue indicates accuracy for canonical views and dark blue indicates accuracy for transformed views. In the left plot, all models share the same ViT-B visual encoder architecture, keeping model complexity constant. In the right plot, all models were pretrained on 400M images, keeping dataset size constant. Human accuracy is shown as red dotted lines, with shaded areas indicating 95% confidence intervals for canonical (light red) and transformed (dark red) views. Other factors, including dataset domain, learning objective, and training setup, are held constant to isolate the effects of dataset size and model complexity on categorization accuracy.

human accuracy under background change conditions, as shown in Figure 12. These results suggest that exposure to large, diverse datasets helps mitigate context dependence.

### Humans and models do not fully agree on which categories are challenging

To assess the alignment between human and model performance at the category level, we analyzed robustness scores for each object category and transformation type (Figure 5). Humans tend to show less variability in robustness across categories than models, indicating more consistent performance. However, more challenging transformations, such as in-depth object rotations, are associated with greater variability across categories for both humans and models. Humans and models also exhibit the highest agreement for in-depth object rotations (Figures 5 and 14), where both struggle with keyboards and perform well on airplanes and bicycles. This suggests that certain intrinsic object properties (e.g., elongated structures or symmetric features) affect recognition across viewpoints in a similar way for both.

For background changes, models and humans show partial agreement, with some object categories, e.g., birds, boats, and airplanes, being more strongly affected by background change than others, e.g., dogs and cats. The former categories may be more tightly associated with specific backgrounds in everyday experience, making them more sensitive to contextual shifts. Among the models, vision-only models exhibit the greatest variability across categories, likely reflecting a stronger reliance on background context for categoriza-

tion. In contrast, humans show higher and more consistent robustness, suggesting that human perception more effectively isolates object identity from its surrounding context.

The greatest divergence occurs in in-plane object rotation, where models struggle more than humans. For example, while humans easily recognize boats across in-plane rotations, models display significant variability in robustness (Figure 5). This suggests that models may depend more on specific object viewpoints or texture cues, whereas humans can generalize more effectively across rotated perspectives.

Beyond overall category difficulty, we also examined category confusions – that is, which object categories humans and models selected when making an error. Across all transformations, human participants showed greater agreement with one another in their category confusions than with any of the models, while models tended to correlate more strongly with each other (Figures 15 and 16). This suggests that although some models approximate human accuracy, they may rely on different representational strategies for categorization.

### Discussion

**Implications for modeling human vision.** Our results highlight the substantial progress that data-rich models have made toward human-like object recognition, while also revealing remaining differences in robustness and error patterns under 3D object transformations. Although top-performing models match or approach human accuracy for moderate transformations like translation and scaling, a performance gap remains for more challenging conditions, especially in-plane and in-

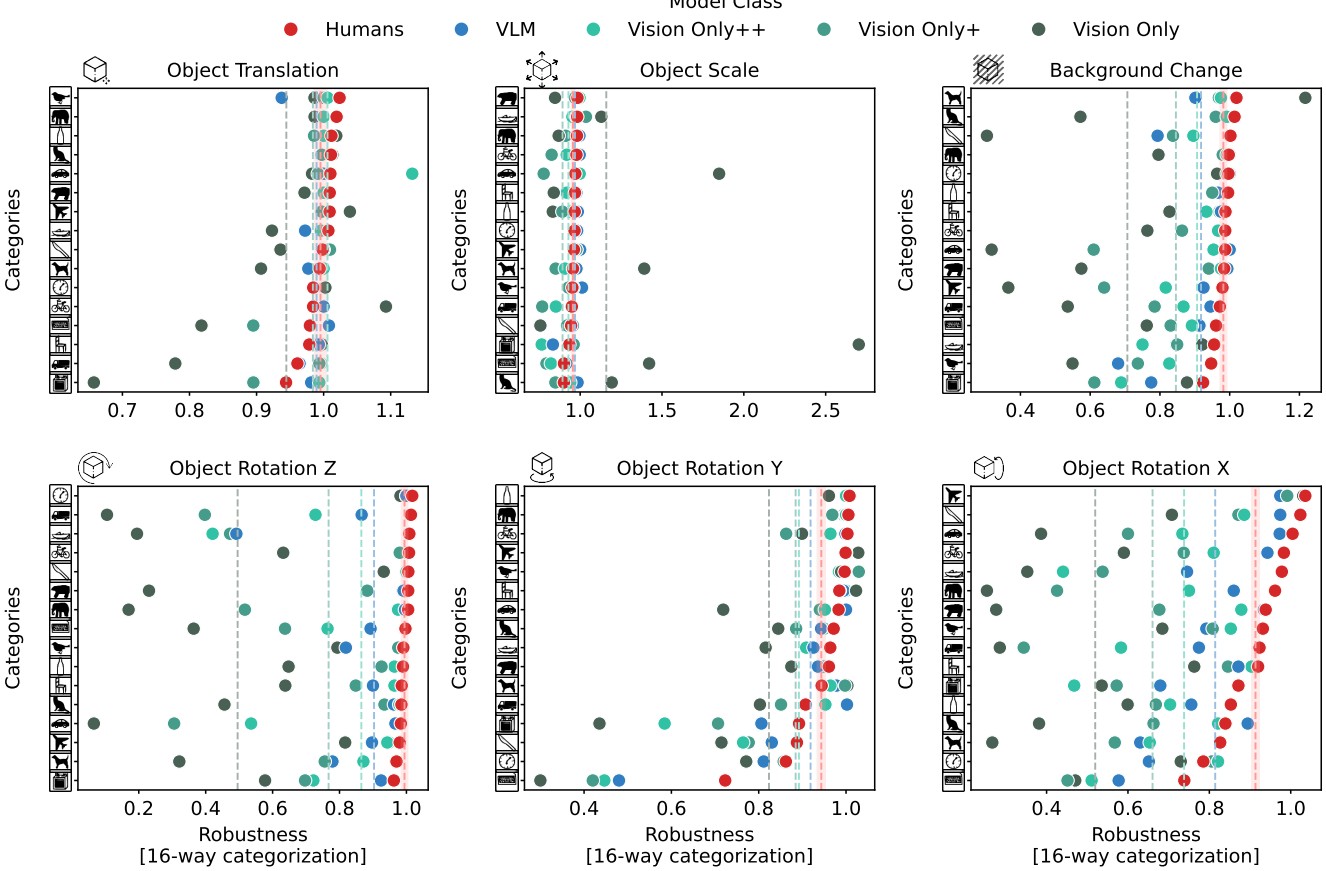

Figure 5: **Robustness of humans and models across different transformations and categories.** Each panel shows robustness scores for 16-way categorization for a specific transformation: object translation, object scaling, background change, in-plane object rotation (z), in-depth object rotation (y), and in-depth object rotation (x). Dotted vertical lines represent the mean robustness for humans (red, with 95% confidence intervals) or models in a family (blues and greens).

depth object rotations, where models exhibit reduced invariance relative to humans. In these regimes, models not only show larger drops in accuracy but also diverge from human error patterns. These discrepancies suggest that current models may rely on different cues or strategies, and that human robustness under 3D transformations likely reflects a combination of visual experience, learning during the task, and mechanisms of viewpoint-invariant representation that models have yet to fully capture.

Importantly, we find that larger training datasets are strongly associated with improved robustness, even when holding architecture and learning objective constant. In particular, our analyses of vision-language models with identical architectures show a clear, monotonic relationship between data scale and accuracy, mirroring trends observed in vision-only models. Model complexity further amplifies these gains, with larger architectures achieving near-human performance. While our study was not designed to fully disentangle these dimensions, the evidence suggests that data richness and architectural scale play a substantial and interacting role in bridging the gap with human perception.

Our results motivate future research aimed at evaluating and developing models with stronger inductive biases, such as explicit 3D representations, temporal context, or embodied interaction (Marr, 2010; Földiák, 1991; Becker, 1992; Smith et al., 2018), which may reflect priors shaped by evolution or experience and support learning of the generative causes of images and scenes (Yuille & Kersten, 2006; Lake et al., 2016). While we focused on models trained to process single static images, recent advances in 3D representation methods – such as point-cloud and generative shape methods and neural radiance fields (NeRFs) – as well as video-based models and agents interacting with 3D environments, offer promising new directions for improving model-to-human alignment in object vision (Mildenhall et al., 2020; Zhang et al., 2023; Z. Chen et al., 2024; Yang et al., 2019). Although these models were not evaluated here, several could, in principle, be adapted to our benchmark using extracted features and linear probing. Their inclusion will be important for testing whether remaining human-model differences reflect architectural limitations,

training exposure, or representational format. At the same time, more challenging tasks, finer-grained behavioral measures, or ambiguous categories could help surface differences that may be obscured in high-performing conditions.

**Implications for theories of human vision.** Our findings also speak to long-standing debates about the nature of human object recognition. The parallel between humans and data-rich models – especially under in-depth rotation – suggests that exposure to a rich and varied set of 2D views may be sufficient to support some aspects of robust recognition. This aligns with view-based theories (Tarr & Bülthoff, 1995; Serre et al., 2007), in which recognition emerges from interpolation or pooling across stored, view-tuned representations. However, the sharp divergence in model performance under in-plane object rotation highlights the limits of view-based training alone. Despite likely limited exposure to such transformations, humans remain nearly invariant, suggesting the involvement of abstract priors, such as assumptions about symmetry or canonical object orientation—consistent with object-centered accounts (Marr, 2010; Biederman, 1987).

Taken together, our results suggest that human 3D object recognition is unlikely to be explained by visual experience alone. A hybrid account, combining extensive exposure to 2D views with structural priors or learned invariances, may better capture the strategies underlying human robustness. Testing this idea will require experiments that better match the visual diet between humans and models, or that systematically vary object-centered structure and symmetry across transformations.

**Limitations.** Our model set spans a wide range of architectures and training regimes, but is limited to deep neural networks that process single static images and support object categorization. This excludes models trained on multiview, temporal, or 3D input, such as video transformers, NeRF-based systems, and point-cloud models. While many of these models were developed for tasks other than categorization, several could be adapted for evaluation on our benchmark using linear probing. Their inclusion in future work will be important for assessing the generality of our findings.

Our behavioral task – a 16-way categorization design using common object categories – was chosen to enable control and large-scale data collection. However, this set captures only a small subset of object categories and may not reflect the full range of visual and semantic variability encountered in everyday vision. Some object categories also varied in visual or semantic variability (e.g., "clock" vs. "bear"), which could limit generalizability to broader recognition contexts. Human accuracy was high across most transformations, which may have reduced sensitivity to fine-grained differences in error consistency. Additionally, humans viewed each object multiple times during the experiment, whereas models are memoryless. Our analysis shows that accuracy for humans increased over blocks, particularly for object rotations where human-model gaps were largest, suggesting that learning or memory effects may contribute to human robustness. Future

work using more diverse, fine-grained, or memory-controlled designs may help clarify the origins of remaining discrepancies.

Lastly, our model comparisons do not fully disentangle the effects of data scale, architecture, and supervision type. While we observe that models with larger training datasets perform better, this pattern co-occurs with changes in architecture complexity and learning objective (e.g., vision-language contrastive learning). More in-depth controlled comparisons along these dimensions will be needed to identify which properties are most critical for achieving human-like robustness to 3D transformations.

**Conclusion.** Despite the progress observed, aligning artificial systems with the flexibility and invariance of human vision remains an open challenge. While our findings shed light on the capabilities of models trained on single static images, we did not evaluate models explicitly designed for 3D object understanding. Incorporating such models will be essential for testing whether richer structural priors and training signals can further narrow the gap. With continued progress and broader model evaluation, models may come closer to capturing the core computational principles that support robust human object recognition across the complexity of the real world.

## Acknowledgements

This work was funded by a Natural Sciences and Engineering Research Council Discovery Grant (RGPIN-2019-06741) and supported by an NSERC Canada Graduate Scholarship – Doctoral (CGS-D).

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

# Appendix

## Stimulus generation: additional details

We constructed a synthetic image dataset by rendering 3D objects under controlled transformations. This process yielded a diverse set of images designed to probe robustness to identity-preserving variation in both humans and models during object categorization. All images were rendered at a resolution of 224 × 224 pixels. For a visual overview, Figure 6 shows all objects in their canonical views, while Figure 7 illustrates the full range of transformations for an example object. Below, we provide a detailed description of the 3D object preprocessing and transformation procedures used to generate the image dataset.

**Preprocessing of 3D objects.** Each 3D object underwent two key preprocessing steps to standardize appearance and positioning:

1. **Uniform scaling**: To ensure consistent object size across the dataset, each 3D object was enclosed in a bounding box defined by its extent along the $x$, $y$, and $z$ axes. The largest dimension of this bounding box was used as a reference for scaling. Models larger than the standard size were scaled down; smaller models were scaled up. Importantly, scaling was applied uniformly across all three dimensions to preserve the object's proportions.

2. **Object centering**: Each object was translated such that its center aligned with the origin of the 3D scene, corresponding to the camera's focal point. This ensured consistent object placement across renderings and minimized spatial biases.

**Identity-preserving transformations.** Stimuli were rendered using Unity, applying the following naturalistic transformations:

- **Object translation**: Objects were shifted from the center of the 3D scene along a polar grid, using orthographic projection to avoid perspective-induced distortion during translation. The translation radius $r$ was incremented in fixed steps of 0.1 from 0 to 0.7 Unity world units, and for each radius, the direction (angle $\theta$) was randomly sampled from a uniform distribution over $[0, 360]°$. This method results in rendered images where objects progressively move away from the center of the image in various random directions. By sampling angles randomly and stepping through radius values uniformly, we ensured a balanced and comprehensive distribution of object placements within the image space, simulating natural spatial variability while keeping most objects within view.

- **Object scaling**: Size variation was introduced by changing the camera's distance from the object, simulating zoom effects under Unity's default perspective projection. The camera was positioned at distances ranging from 0.3 to 3 Unity world units, with the canonical view defined at a distance of 1 unit. Decreasing the camera distance below 1 unit resulted in zooming in, while increasing it led to zooming out. As a consequence, the rendered object size in the image varied from 200% to 12.5% of the canonical size, where size was defined as the maximum of the object's height or width in image space. We used eight scale levels: 12.5%, 25%, 50%, 75%, 100% (canonical), 125%, 150%, and 200%.

- **Object rotation**: Objects were placed with their centers at the world origin and rotated a full 360° around each of the three world axes, one axis at a time. We applied in-plane rotations around the z-axis, in-depth rotations from side to side around the y-axis, and in-depth rotations from top to bottom around the x-axis. For each axis, we captured images at 45° increments, resulting in eight views per axis, including the canonical (unrotated) view. Images were rendered using Unity's default perspective projection, from a fixed camera positioned 1 Unity world unit from the object, with 0° elevation from the xz-plane and 0° azimuth from the yz-plane.

- **Background change**: Background variation was introduced using a set of six distinct images per object category. We used a gray default background with RGB values of (0.485, 0.456, 0.406) Ollikka et al. (2025), matching the average pixel color of ImageNet images. This background was applied by default in all other augmentations. To introduce contextual variation, the gray background was replaced with either a colored 1/f noise control or one of six naturalistic scenes selected for each category. These naturalistic scenes were divided into two types – three congruent and three incongruent – based on their typical relevance to each object category. For instance, if the object was a car, congruent backgrounds included highways or outdoor scenes typical for cars, while incongruent backgrounds featured indoor environments. Unlike the 3D transformations, background change leaves the object unchanged and varies only the surrounding context, providing a complementary test of robustness to identity-preserving variations. The 1/f noise background was included as a control to introduce structured variation without recognizable content, in contrast to the scene-like content of naturalistic backgrounds.

## Human experiment: additional details

After providing informed consent, participants completed the experiment remotely on their personal computers. We designed the experiment in PsychoPy Peirce et al. (2019) and ran it online via Pavlovia (Open Science Tools, Nottingham, UK). To estimate screen size and viewing distance, we used a credit card scaling procedure and blind spot estimation method, respectively (Brascamp, 2021). Participants were given detailed instructions on the categorization task, including how to respond using a mouse click, and were advised to

familiarize themselves with the 16 object categories presented on the response screen. They were encouraged to answer as accurately as possible and to rely on their best judgment when uncertain.

**Practice blocks.** The experiment began with two practice blocks of 128 trials each, using randomly selected ImageNet validation images, i.e., photos of real-world objects belonging to one of the 16 categories. These blocks served two purposes: to familiarize participants with the task and to identify inattentive participants. The first block included visual feedback (a green checkmark for correct responses, a red cross for incorrect or missed responses, and a highlight on the correct category), while the second block did not. Participants who scored below 80% accuracy on the second block were excluded from the study and were not invited to continue with the main experiment. Those who passed the accuracy threshold completed a third practice block of 128 trials, using synthetic images generated via our stimulus generation pipeline, which applied naturalistic transformations to 3D object models using the Unity game engine. These images were drawn from the same distribution as those used in the main experiment but featured a separate set of 3D object exemplars not included in the test set (see Figure 6); this block also included visual feedback.

For ImageNet blocks, we used Python to preprocess the images. From the pool of ImageNet images representing the 16 basic-level categories, we excluded grayscale images (approximately $1\%$), images smaller than 256 × 256 pixels, and images depicting multiple objects from the target categories. Each image was cropped to the largest possible central square and then resized to 224 × 224 pixels using the PIL.Image.thumbnail((224, 224)) method.

**Experimental blocks.** The main experiment consisted of 10 blocks of 128 trials each, using synthetic images from the same generation pipeline introduced in the third practice block. These images were produced by applying naturalistic transformations to 3D object models using the Unity game engine (see Figures 6 and 7). To help maintain participant engagement throughout the session, a performance summary was shown at the end of each block. After each block, participants were given a mandatory 30-second break and could resume the task at their own pace with a mouse click. The entire experiment lasted approximately 1.5 hours (see Figure 8 for an overview of the experimental design and block structure).

**Risks.** There are no known risks associated with the computer-based visual tasks used in this study. Participants were informed that they could take breaks between blocks or terminate the session at any time if they felt tired or uncomfortable.

# Categories

Exemplars

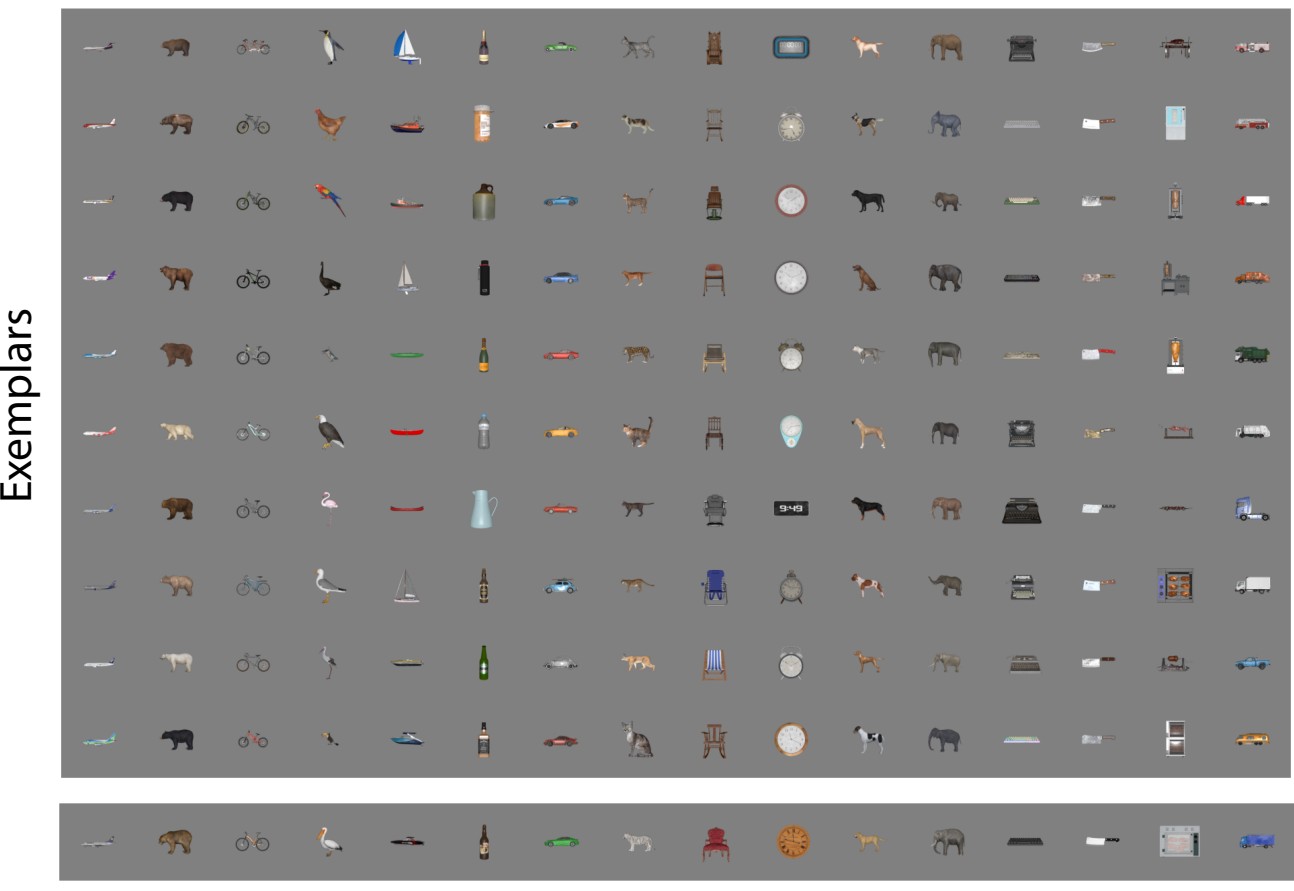

Figure 6: **Overview of 3D object models used in the ORBIT benchmark.** Objects are shown in their canonical view, organized by category (columns) and exemplar (rows). The top panel shows objects used in both human and model evaluation; the bottom panel shows objects used during practice in the human experiment.

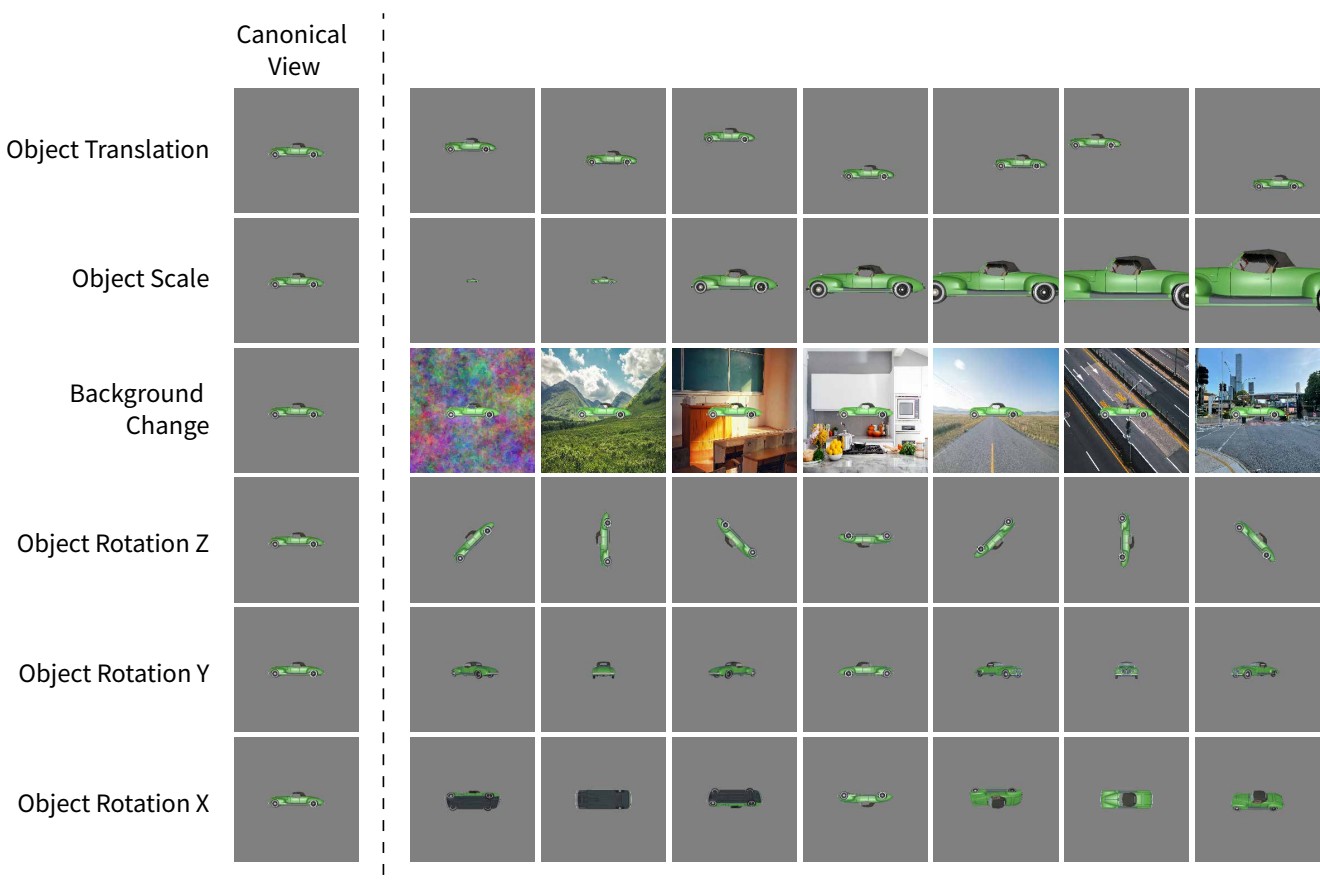

Figure 7: **Illustration of the six 3D transformations applied to objects in the ORBIT benchmark.** The leftmost column shows the canonical view, while subsequent columns display variations introduced by each transformation: object translation (shifting object position within the frame), object scale (changing object size), background change (altering the background while keeping the object unchanged), object rotation z (in-plane rotation), object rotation y (rotation around the vertical axis), and object rotation x (rotation around the horizontal axis).

A   Trial (~2 secs)

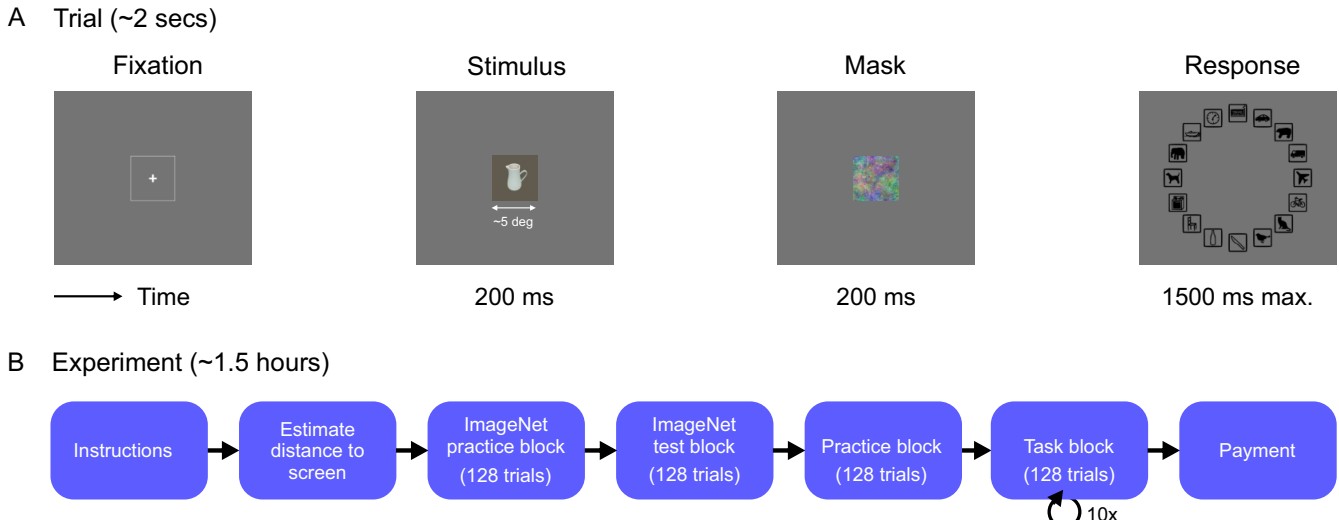

B   Experiment (~1.5 hours)

Figure 8: **Overview of the human experiment.** Participants were instructed to categorize object images in a forced-choice paradigm. (A) Participants initiated each trial by clicking on the fixation cross, which started the presentation of a stimulus followed by a noise mask, each presented for 200 ms, and a response screen that was presented for 1500 ms or until the participant clicked on one of the category icons, whichever came earlier. Stimuli and masks were presented at 5 degrees of visual angle. (B) Schematic of the experiment, which consisted of three practice blocks and 10 experimental task blocks. Unknown to the participants, the second practice block served as a test: only participants with an accuracy of 80 percent or higher were included in the experiment and invited to continue.

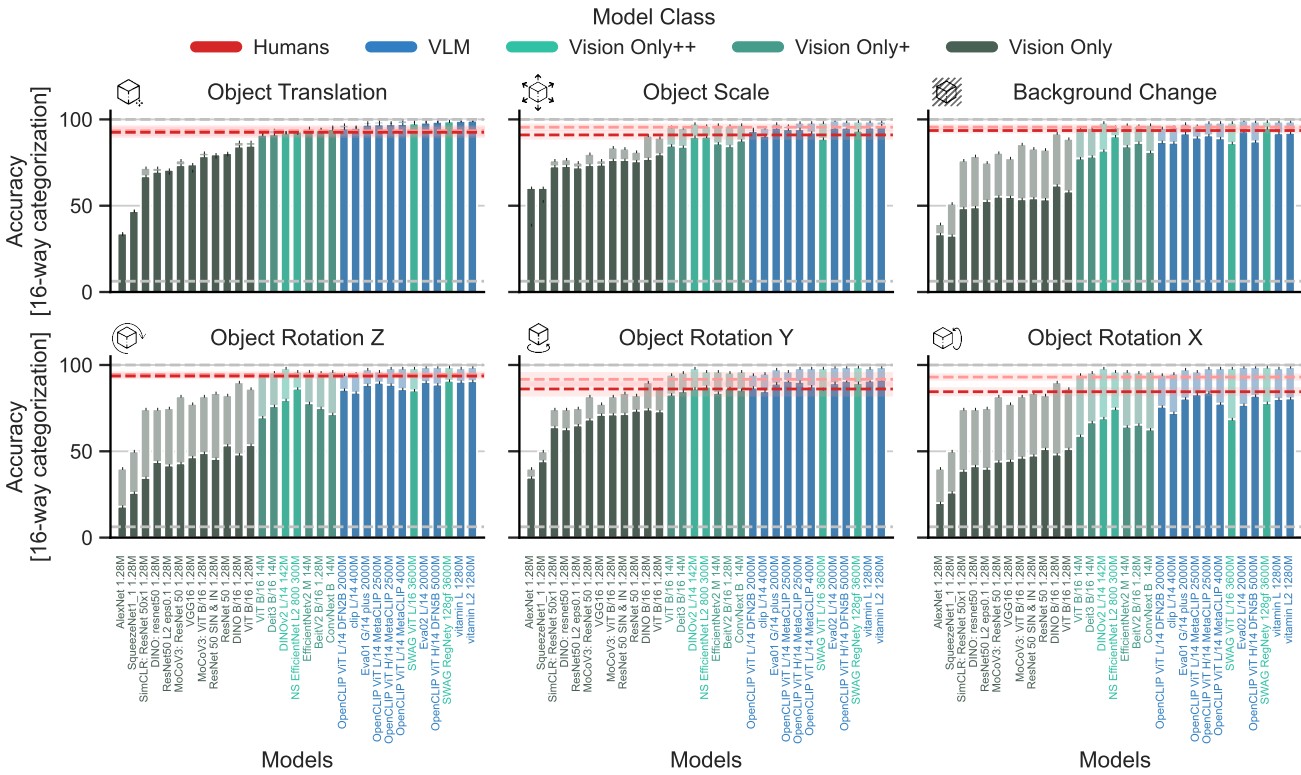

Figure 9: **Performance across different transformations in a 16-way object categorization task.** Each panel shows accuracy for a specific transformation: object translation, object scale, background change, object rotation z (in-plane), object rotation y (front-back), and object rotation x (top-bottom). Human accuracy is represented by the red dashed lines (with 95% confidence intervals), while the gray dashed lines indicate chance-level performance. Lighter colors indicate accuracy for canonical views; darker colors indicate accuracy for transformed views.

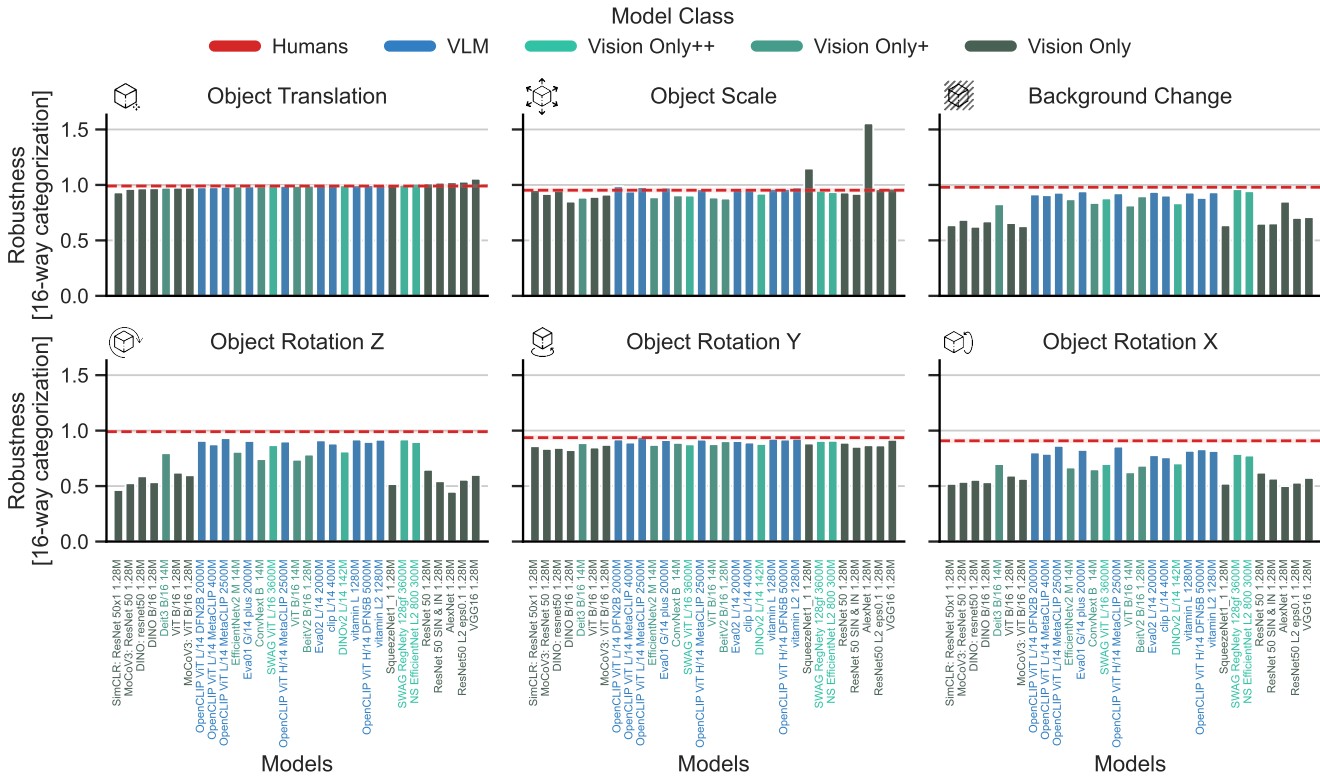

Figure 10: **Robustness across different transformations in a 16-way object categorization task.** Each panel shows robustness for a specific transformation: object translation, object scale, background change, object rotation z (in-plane), object rotation y (front-back), and object rotation x (top-bottom). Human robustness is represented by the red dashed line (with 95% confidence intervals).

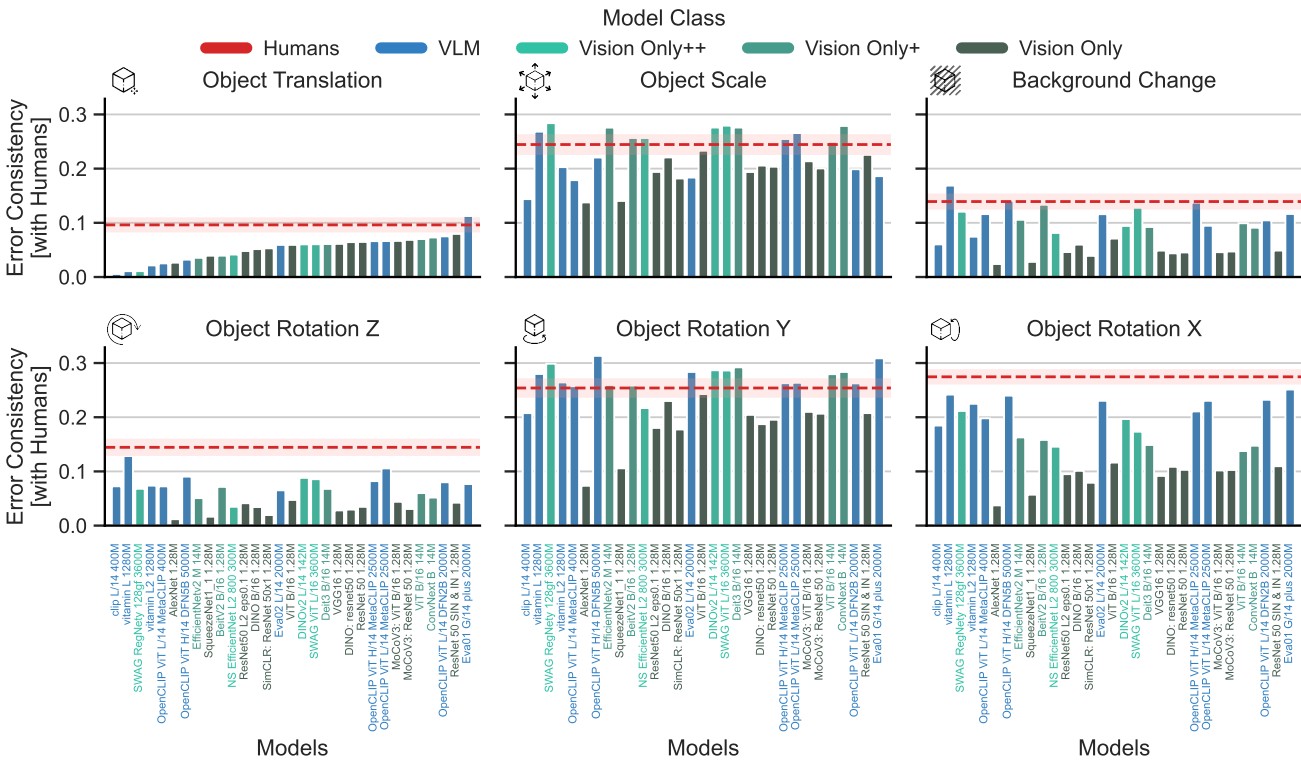

Figure 11: **Model-to-human error consistency across different transformations in a 16-way object categorization task.** Error consistency measures the likelihood that models make the same classification errors as human participants when objects undergo transformations. Each panel shows error consistency for a specific transformation: object translation, object scale, background change, object rotation z (in-plane), object rotation y (front-back), and object rotation x (top-bottom). Human error consistency is represented by the red dashed lines (with 95% confidence intervals).

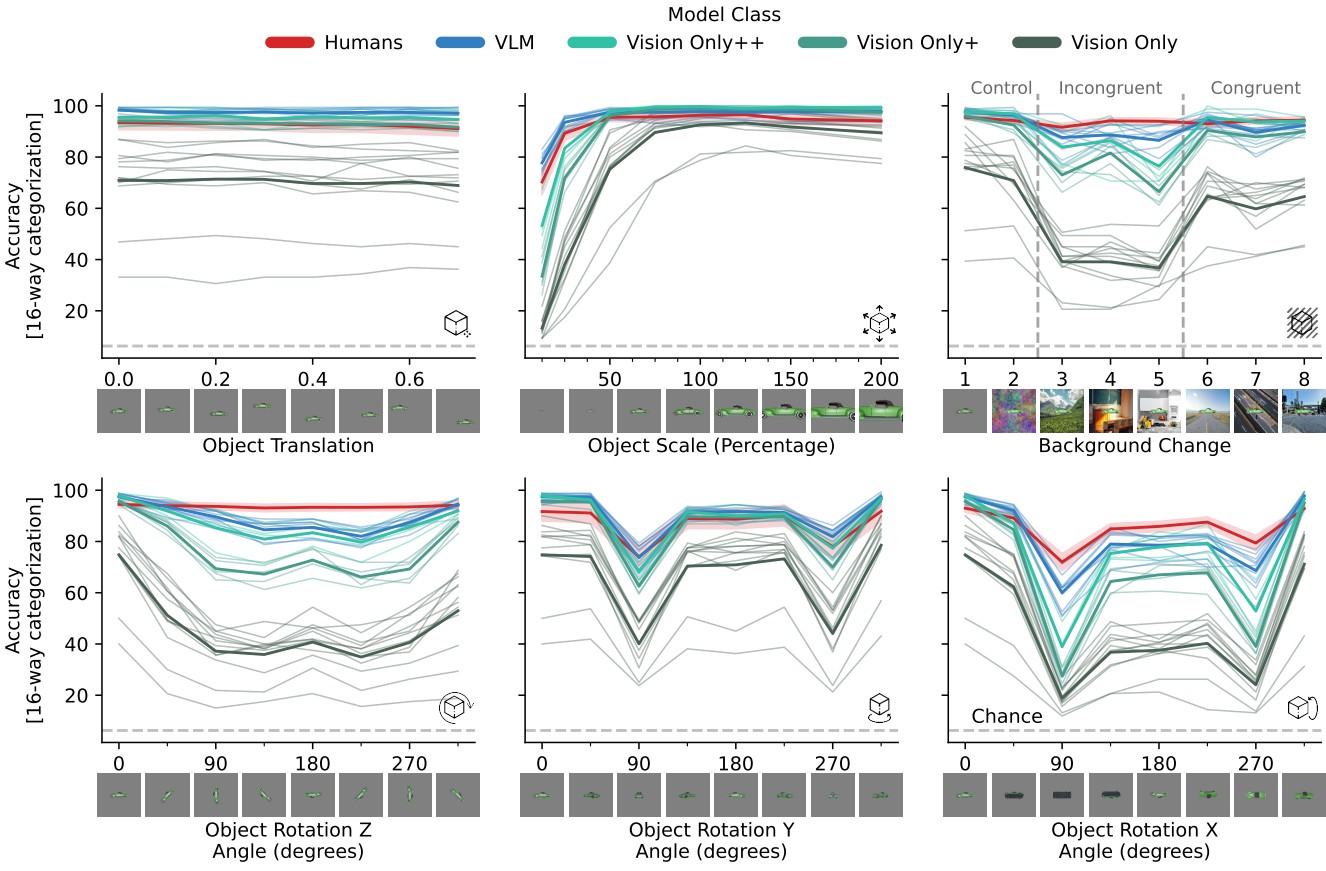

Figure 12: **Human and model performance as a function of transformation level across different transformations.** Each panel shows accuracy at 16-way categorization for a specific transformation: object translation, object scale, background change, in-plane object rotation (z), in-depth object rotation (y), and in-depth object rotation (x). The dashed gray line represents chance performance. The thick colored lines show mean performance across humans (red, with 95% confidence intervals) or models in a family (blues and greens). The thin colored lines show performance for individual models.

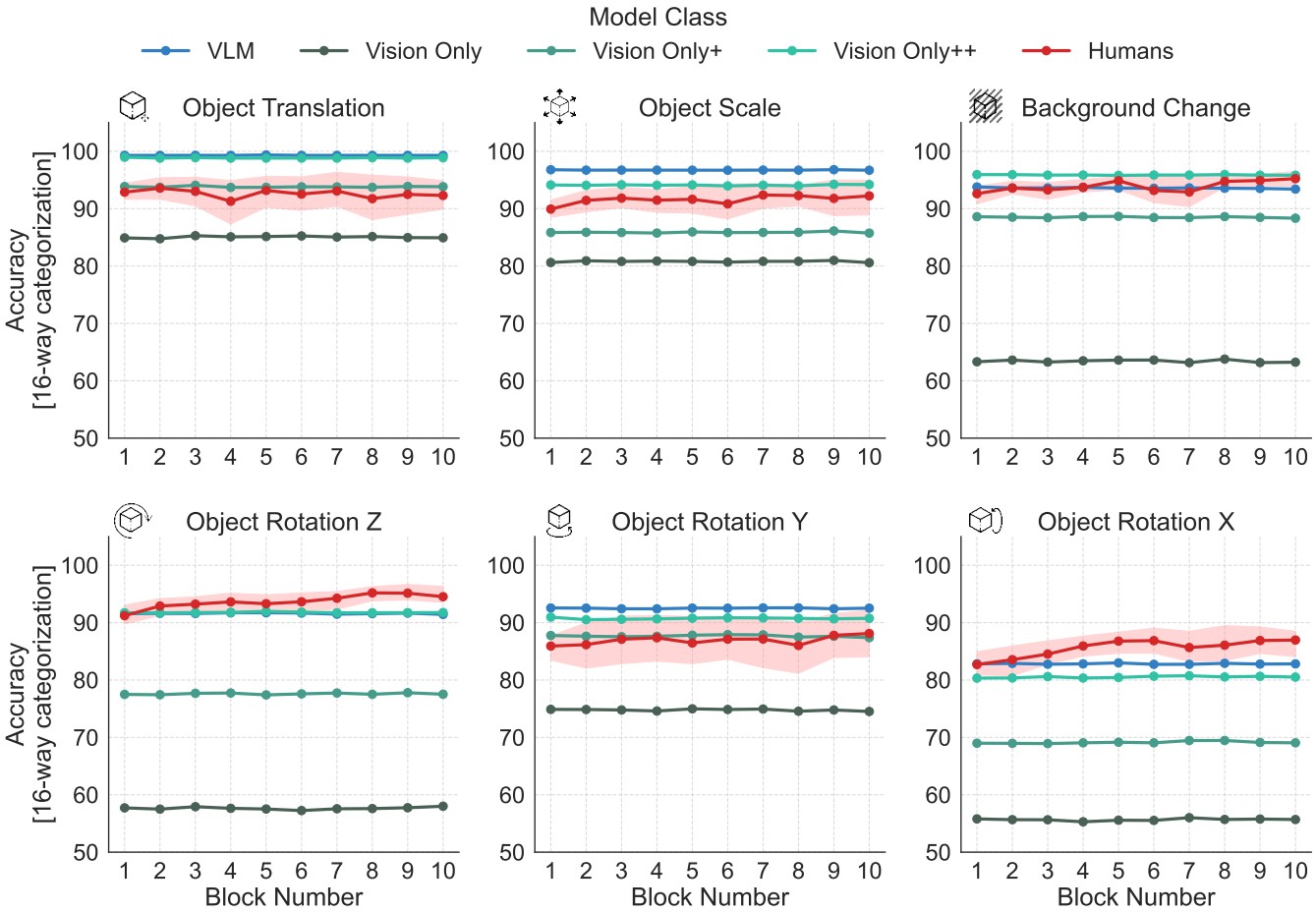

Figure 13: **Human and model performance as a function of experimental test block.** The colored lines show accuracy at the task (canonical and transformed views together) for humans (red, with 95% confidence intervals) and the best-performing model of each family (blues and greens).

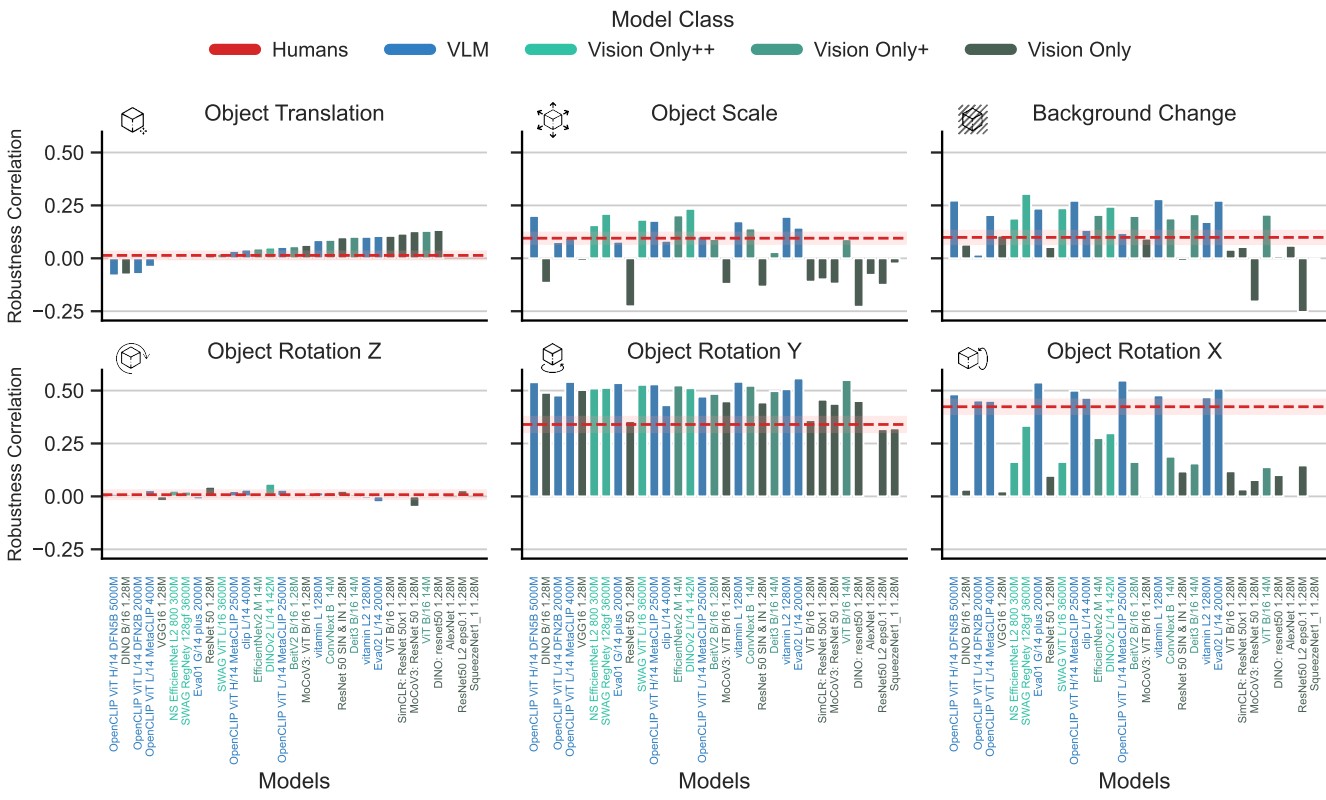

Figure 14: **Model-to-human category-level alignment across different transformations.** We computed robustness scores across object categories (16-element vectors) and correlated these scores between individual models and individual humans. Bars show the average model-to-human correlations (Pearson r) for individual models. The red dashed lines indicate average human-to-human correlations, with shaded areas indicating the 95% confidence intervals. Stronger positive correlations indicate stronger agreement on which categories are challenging.

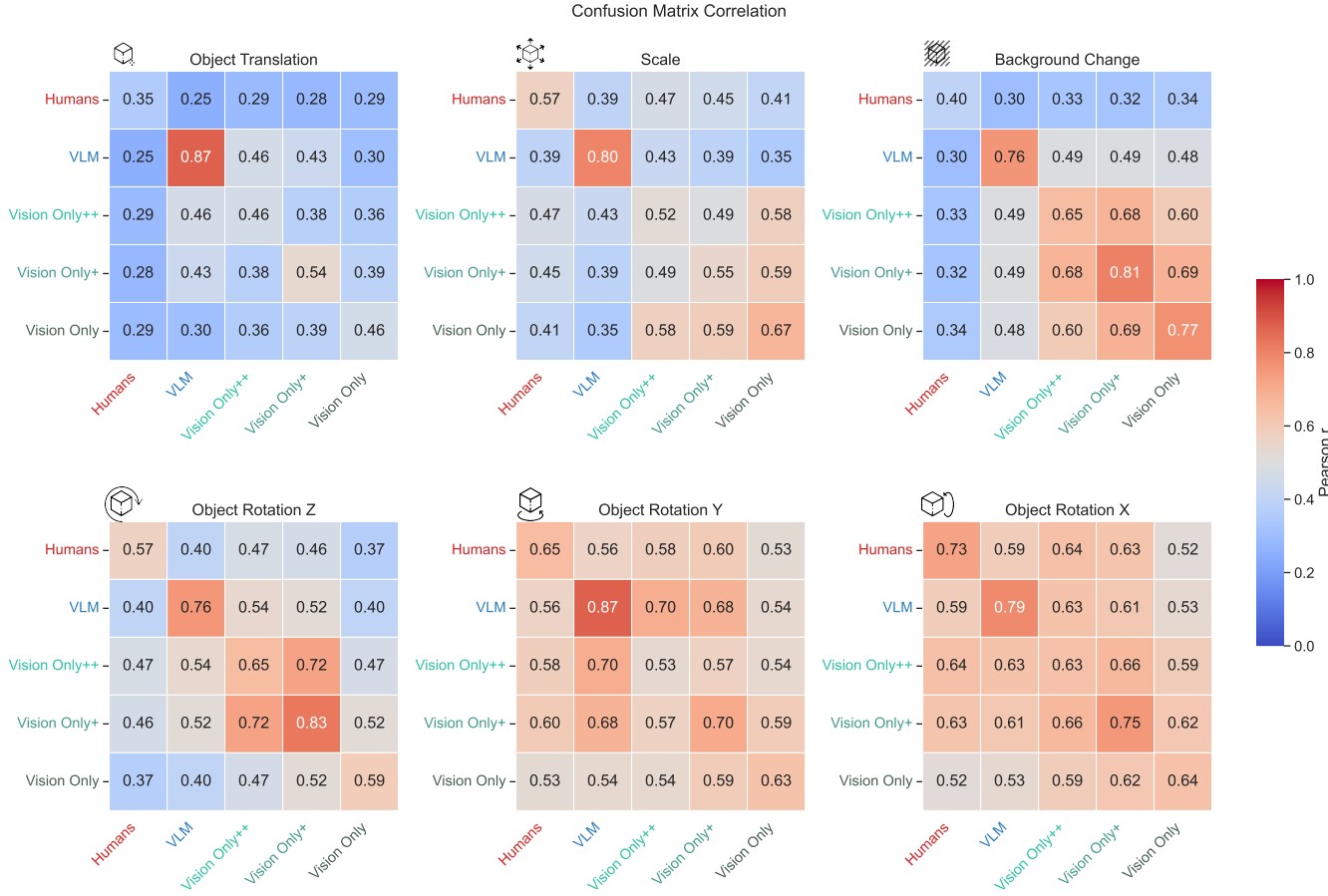

Figure 15: **Alignment on category confusions, within and between humans and models, across different transformations.** We computed category confusion matrices for models and humans, whose off-diagonal entries indicate how often each true category is confused with other categories. We correlated the off-diagonal entries of these matrices between individual models and individual humans, and summarized the results – averaging across humans and across models within a family – in the displayed correlation matrices. Stronger positive correlations indicate stronger agreement on category confusions.

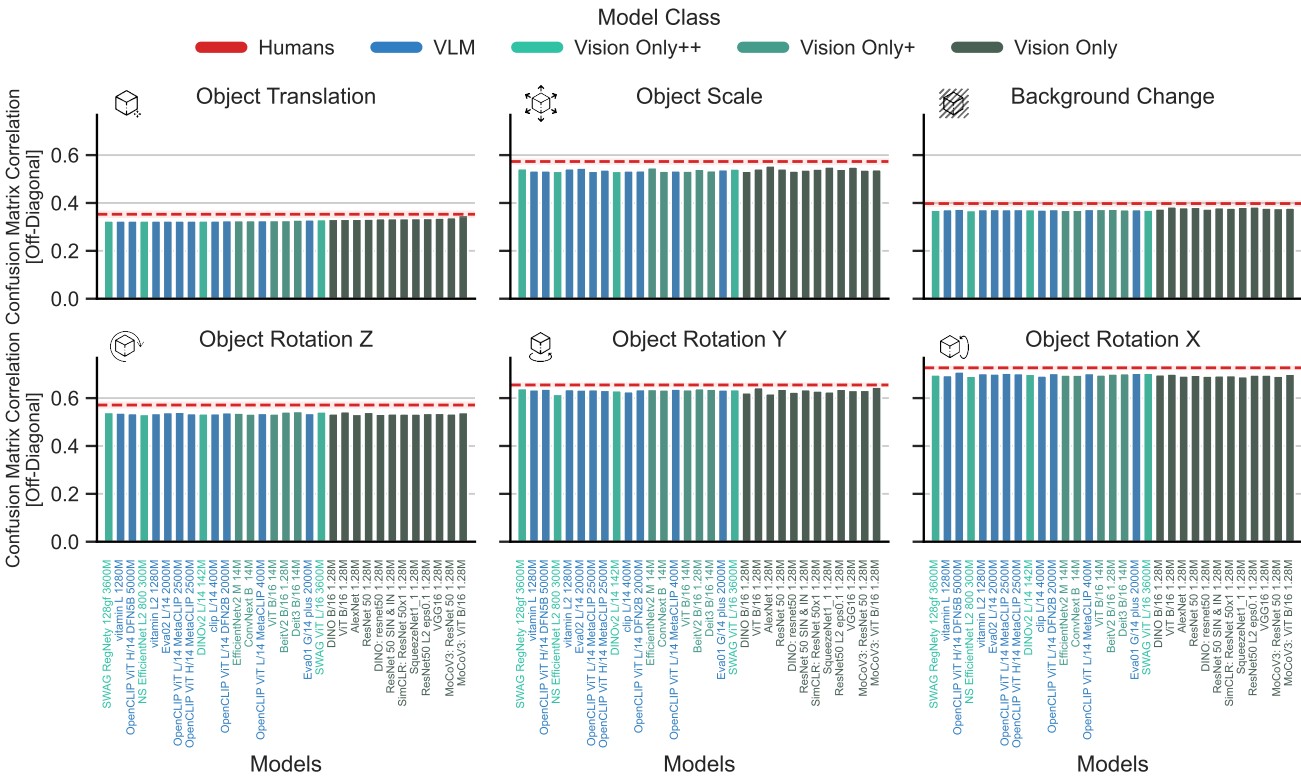

Figure 16: **Model-to-human alignment on category confusions across different transformations.** We computed category confusion matrices for models and humans, whose off-diagonal entries indicate how often each true category is confused with other categories. We correlated the off-diagonal entries of these matrices between individual models and individual humans. We show alignment between humans and individual models (first row/column of the matrices shown in Figure 15, but not averaged across models within a family). Bars show the average model-to-human correlations (Pearson r) for individual models. The red dashed line indicates the average human-to-human correlation, with shaded areas indicating the 95% confidence interval. Stronger positive correlations indicate stronger agreement on category confusions.

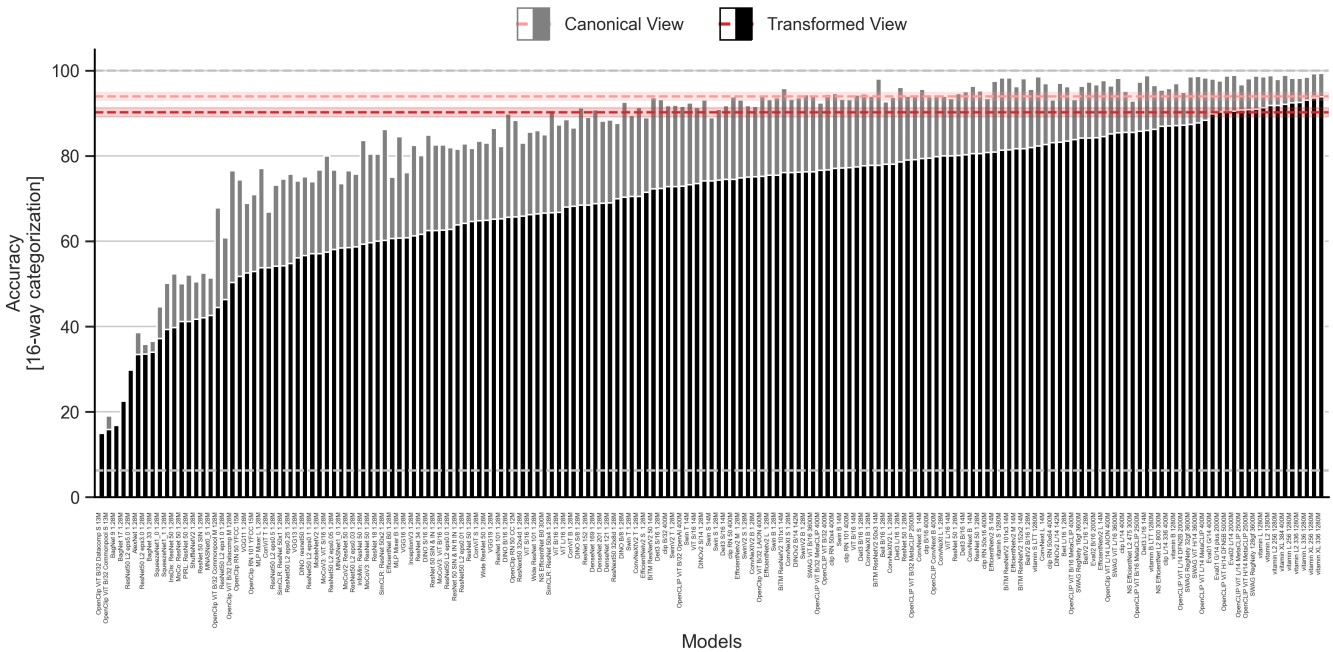

Figure 17: **Performance on the 16-way object categorization task for the full model set (n = 169 ).** Performance was evaluated across canonical (gray bars) and transformed (black bars) views, and averaged across transformations. Each bar represents a distinct computational model ordered by their performance on transformed views. Human performance is indicated by the red dashed line (mean accuracy) with the surrounding shaded area denoting the 95% confidence interval.

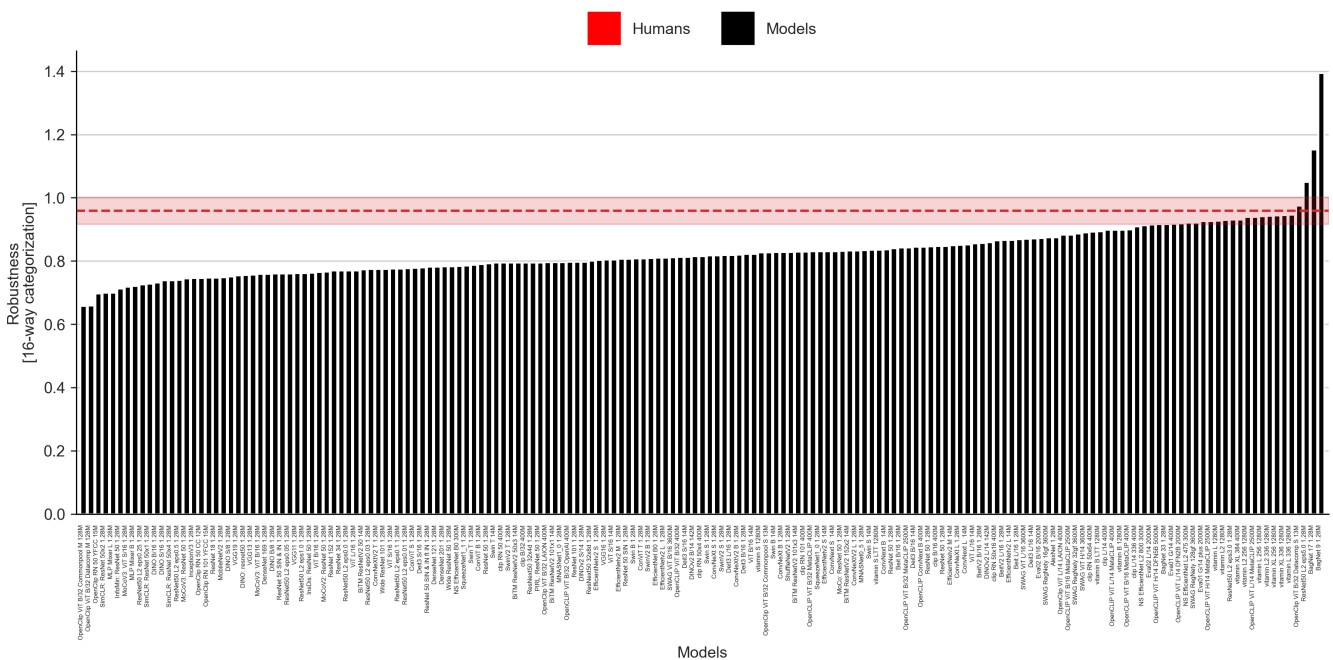

Figure 18: **Robustness on the 16-way object categorization task for the full model set (n = 169 ).** Each bar corresponds to an individual model, ordered by robustness scores. Human robustness is represented by the red dashed line, with shaded regions indicating the 95% confidence interval. Values close to 1 indicate high robustness, with transformed-view accuracy comparable to canonical-view accuracy.

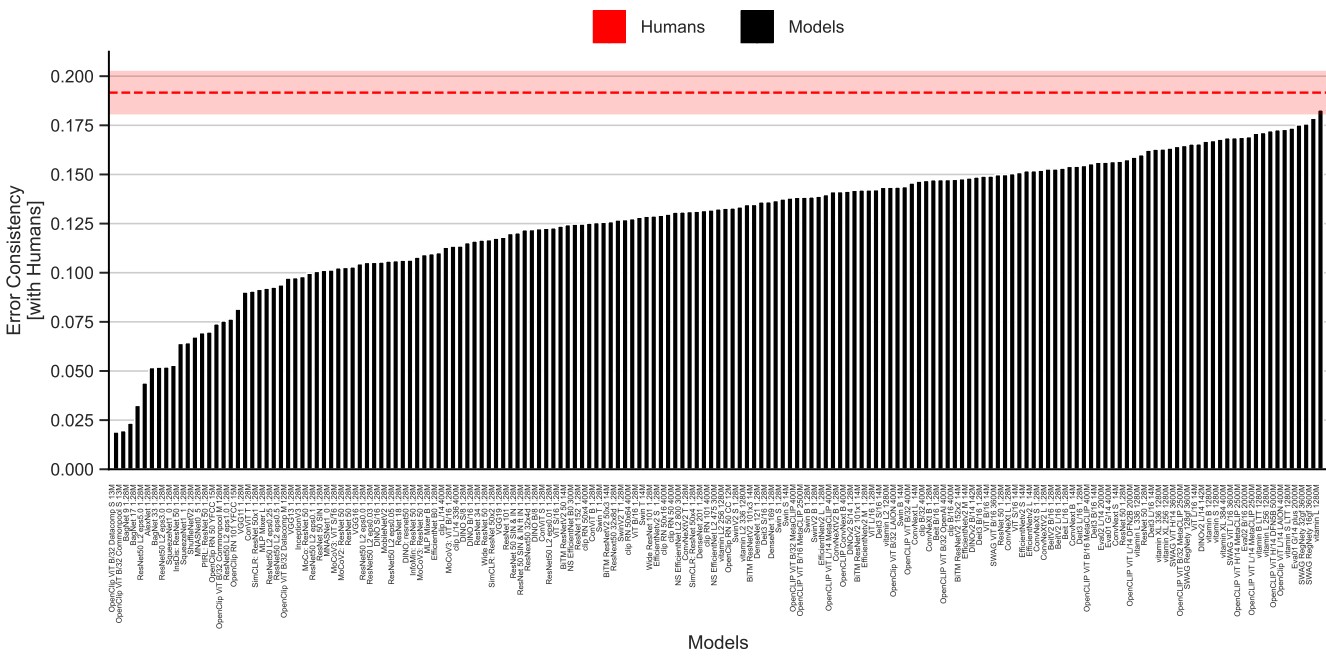

Figure 19: **Model-to-human error consistency on the 16-way object categorization task for the full model set (n = 169 ).** Error consistency measures the likelihood that models make the same classification errors as human participants when objects undergo transformations. Error consistency was averaged across transformations. Human error consistency is represented by the red dashed line with the surrounding shaded area denoting the 95% confidence interval.

Table 2: **List of all 169 Neural Network Models Evaluated.**

| Model |
| --- |
| CLIP ViT L/14 400M (Schuhmann et al., 2022; Radford et al., 2021) |
| CLIP-A ViT H/14 (Schuhmann et al., 2022; Radford et al., 2021) |
| CLIP RN50 YFCC15M (Schuhmann et al., 2022; Radford et al., 2021) |
| CLIP RN50 CC12M (Schuhmann et al., 2022; Radford et al., 2021) |
| CLIP RN101 YFCC15M (Schuhmann et al., 2022; Radford et al., 2021) |
| CLIP ViT B/32 DataComp-S (Schuhmann et al., 2022; Radford et al., 2021) |
| CLIP ViT B/32 DataComp-M (Schuhmann et al., 2022; Radford et al., 2021) |
| CLIP ViT B/32 CommonPool-S (Schuhmann et al., 2022; Radford et al., 2021) |
| CLIP ViT B/32 CommonPool-M (Schuhmann et al., 2022; Radford et al., 2021) |
| CLIP ConvNeXt Base (Schuhmann et al., 2022; Radford et al., 2021) |
| CLIP ViT B/32 400M (Schuhmann et al., 2022; Radford et al., 2021) |
| CLIP ViT H/14 DFN5B (Schuhmann et al., 2022; Radford et al., 2021) |
| CLIP ViT H/14 MetaCLIP (Schuhmann et al., 2022; Radford et al., 2021) |
| CLIP ViT L/14 MetaCLIP (Schuhmann et al., 2022; Radford et al., 2021) |
| CLIP ViT L/14 MetaCLIP-400 (Schuhmann et al., 2022; Radford et al., 2021) |
| CLIP ViT L/14 DFN2B (Schuhmann et al., 2022; Radford et al., 2021) |
| CLIP ViT B/16 MetaCLIP (Schuhmann et al., 2022; Radford et al., 2021) |
| CLIP ViT B/32 QuickGELU-OpenAI (Schuhmann et al., 2022; Radford et al., 2021) |
| CLIP ViT B/32 QuickGELU-400M (Schuhmann et al., 2022; Radford et al., 2021) |
| CLIP ViT B/32 MetaCLIP-400M (Schuhmann et al., 2022; Radford et al., 2021) |
| CLIP ViT B/16 MetaCLIP-400M (Schuhmann et al., 2022; Radford et al., 2021) |
| CLIP ViT B/32 MetaCLIP (Schuhmann et al., 2022; Radford et al., 2021) |
| SigLIP ViT B/16 Zhai et al. (2023) |
| Vitamin L 336 1B J. Chen et al. (2024) |
| Vitamin L 256 1B J. Chen et al. (2024) |
| Vitamin XL 384 1B J. Chen et al. (2024) |
| Vitamin XL 336 1B J. Chen et al. (2024) |
| Vitamin XL 256 1B J. Chen et al. (2024) |
| Vitamin L2 336 1B J. Chen et al. (2024) |
| Vitamin L2 256 1B J. Chen et al. (2024) |

Table 3: **List of all 169 Neural Network Models Evaluated.**

| Model |
| --- |
| Vitamin L2 1B J. Chen et al. (2024) |
| Vitamin L 1B J. Chen et al. (2024) |
| Vitamin B LTT 1B J. Chen et al. (2024) |
| Vitamin B 1B J. Chen et al. (2024) |
| Vitamin S LTT 1B J. Chen et al. (2024) |
| Vitamin S 1B J. Chen et al. (2024) |
| EVA01 ViT G/14 400M Fang et al. (2022) |
| EVA01 ViT G/14 Plus 2B Fang et al. (2022) |
| EVA02 ViT L/14 2B Fang et al. (2022) |
| EVA02 ViT B/16 2B Fang et al. (2022) |
| CLIP L/14 336 (Radford et al., 2021) |
| CLIP RN50x64 (Radford et al., 2021) |
| CLIP RN50x16 (Radford et al., 2021) |
| CLIP RN50x4 (Radford et al., 2021) |
| CLIP L/14 (Radford et al., 2021) |
| CLIP RN101 (Radford et al., 2021) |
| CLIP B/32 (Radford et al., 2021) |
| CLIP B/16 (Radford et al., 2021) |
| CLIP RN50 (Radford et al., 2021) |
| BEiT Base Patch16 224 Bao et al. (2022) |
| BEiT Large Patch16 224 Bao et al. (2022) |
| BEiTv2 Base Patch16 224 Peng et al. (2022) |
| BEiTv2 Large Patch16 224 Peng et al. (2022) |
| ConvNeXt Small Liu, Mao, et al. (2022) |
| ConvNeXt Large Liu, Mao, et al. (2022) |
| ConvNeXt Base Liu, Mao, et al. (2022) |
| ConvNeXt Base ImageNet-22K FT1K Liu, Mao, et al. (2022) |
| ConvNeXt Large ImageNet-22K FT1K Liu, Mao, et al. (2022) |
| ConvNeXt Small ImageNet-22K FT1K Liu, Mao, et al. (2022) |
| ConvNeXt Large v2 |
| ConvNeXt Base v2 |
| ConvNeXt Tiny v2 |
| MAE ViT Huge FT1K He et al. (2021) |
| MAE ViT Large FT1K He et al. (2021) |
| MAE ViT Base FT1K He et al. (2021) |
| MAE ViT Base He et al. (2021) |
| MAE ViT Large He et al. (2021) |
| MAE ViT Huge He et al. (2021) |
| EfficientNet B0 Q. Xie et al. (2020) |
| EfficientNet B0 Noisy Student Q. Xie et al. (2020) |
| EfficientNet L2 Noisy Student Q. Xie et al. (2020) |
| EfficientNet L2 Noisy Student 475 Q. Xie et al. (2020) |
| EfficientNetV2 L ImageNet-21K FT1K Tan & Le (2021) |
| EfficientNetV2 M ImageNet-21K FT1K Tan & Le (2021) |
| EfficientNetV2 S ImageNet-21K FT1K Tan & Le (2021) |
| EfficientNetV2 S Tan & Le (2021) |
| EfficientNetV2 M Tan & Le (2021) |
| EfficientNetV2 L Tan & Le (2021) |
| SWAG ViT B/16 ImageNet-1K Singh et al. (2022) |

Table 4: **List of all 169 Neural Network Models Evaluated.**

| Model |
| --- |
| SWAG ViT L/16 ImageNet-1K Singh et al. (2022) |
| SWAG ViT H/14 ImageNet-1K Singh et al. (2022) |
| SWAG RegNetY 16GF ImageNet-1K Singh et al. (2022) |
| SWAG RegNetY 32GF ImageNet-1K Singh et al. (2022) |
| SWAG RegNetY 128GF ImageNet-1K Singh et al. (2022) |
| BiT-M ResNetV2 50x1 ImageNet-21K Kolesnikov et al. (2020) |
| BiT-M ResNetV2 50x3 ImageNet-21K Kolesnikov et al. (2020) |
| BiT-M ResNetV2 101x1 ImageNet-21K Kolesnikov et al. (2020) |
| BiT-M ResNetV2 101x3 ImageNet-21K Kolesnikov et al. (2020) |
| BiT-M ResNetV2 152x2 ImageNet-21K Kolesnikov et al. (2020) |
| InsDis Caron et al. (2021) |
| MoCo He et al. (2020) |
| PIRL Misra & Maaten (2019) |
| MoCoV2 X. Chen et al. (2020) |
| MoCoV3 ResNet50 X. Chen et al. (2021) |
| InfoMin Y. Chen et al. (2023) |
| MoCoV3 ViT-S X. Chen et al. (2021) |
| MoCoV3 ViT-B X. Chen et al. (2021) |
| ResNet50 L2 eps0 Salman et al. (2020) |
| ResNet50 L2 eps0.25 Salman et al. (2020) |
| ResNet50 L2 eps0.1 Salman et al. (2020) |
| ResNet50 L2 eps0.05 Salman et al. (2020) |
| ResNet50 L2 eps0.03 Salman et al. (2020) |
| ResNet50 L2 eps0.01 Salman et al. (2020) |
| ResNet50 L2 eps0.5 Salman et al. (2020) |
| ResNet50 L2 eps1 Salman et al. (2020) |
| ResNet50 L2 eps3 Salman et al. (2020) |
| ResNet50 L2 eps5 Salman et al. (2020) |
| SwinV2 Tiny Window8 256 Liu, Hu, et al. (2022) |
| SwinV2 Small Window8 256 Liu, Hu, et al. (2022) |
| SwinV2 Base Window8 256 Liu, Hu, et al. (2022) |
| Swin Tiny Window7 FT21K Liu et al. (2021) |
| Swin Small Window7 FT21K Liu et al. (2021) |
| Swin Base Window7 FT21K Liu et al. (2021) |
| Swin Tiny Window7 Liu et al. (2021) |
| Swin Small Window7 Liu et al. (2021) |
| Swin Base Window7 Liu et al. (2021) |
| MLP-Mixer B/16 224 Tolstikhin et al. (2021) |
| MLP-Mixer L/16 224 Tolstikhin et al. (2021) |
| ResNet50 SWSL Yalniz et al. (2019) |
| ResNeXt50 32x4d SWSL Yalniz et al. (2019) |
| ResNeXt101 32x16d SWSL Yalniz et al. (2019) |
| ConViT Base d'Ascoli et al. (2022) |
| ConViT Small d'Ascoli et al. (2022) |
| ConViT Tiny d'Ascoli et al. (2022) |
| DINOv2 ViT S/14 Oquab et al. (2024) |
| DINOv2 ViT B/14 Oquab et al. (2024) |
| DINOv2 ViT G/14 Oquab et al. (2024) |

Table 5: **List of all 169 Neural Network Models Evaluated.**

| Model |
| --- |
| DINOv2 ViT L/14 Oquab et al. (2024) |
| DINO ViT S/16 Caron et al. (2021) |
| DINO ViT S/8 Caron et al. (2021) |
| DINO ViT B/8 Caron et al. (2021) |
| DINO ViT B/16 Caron et al. (2021) |
| DINO ResNet50 Caron et al. (2021) |
| ViT S/16 Dosovitskiy et al. (2021) |
| ViT B/16 Dosovitskiy et al. (2021) |
| ViT L/16 224 Dosovitskiy et al. (2021) |
| ViT L/16 ImageNet-21K Dosovitskiy et al. (2021) |
| ViT B/16 ImageNet-21K Dosovitskiy et al. (2021) |
| ViT S/16 ImageNet-21K Dosovitskiy et al. (2021) |
| DeiT3 S/16 224 Touvron et al. (2022) |
| DeiT3 B/16 224 Touvron et al. (2022) |
| DeiT3 L/16 224 Touvron et al. (2022) |
| DeiT3 B/16 FT1K Touvron et al. (2022) |
| DeiT3 L/16 FT1K Touvron et al. (2022) |
| DeiT3 S/16 FT1K Touvron et al. (2022) |
| AlexNet Krizhevsky et al. (2012) |
| VGG11 BN Simonyan & Zisserman (2015) |
| VGG13 BN Simonyan & Zisserman (2015) |
| VGG16 BN Simonyan & Zisserman (2015) |
| VGG19 BN Simonyan & Zisserman (2015) |
| SqueezeNet 1.0 Iandola et al. (2016) |
| SqueezeNet 1.1 Iandola et al. (2016) |
| DenseNet121 Huang et al. (2018) |
| DenseNet169 Huang et al. (2018) |
| DenseNet201 Huang et al. (2018) |
| Inception v3 |
| ResNet18 He et al. (2015) |
| ResNet34 He et al. (2015) |
| ResNet50 He et al. (2015) |
| ResNet101 He et al. (2015) |
| ResNet152 He et al. (2015) |
| ShuffleNet v2 x0.5 Ma et al. (2018) |
| MobileNet v2 Sandler et al. (2019) |
| ResNeXt50 32x4d S. Xie et al. (2017) |
| ResNeXt101 32x8d S. Xie et al. (2017) |
| Wide ResNet50-2 |
| Wide ResNet101-2 |
| MNASNet 0.5 Tan et al. (2019) |
| MNASNet 1.0 Tan et al. (2019) |
| BagNet9 Brendel & Bethge (2019) |
| BagNet17 Brendel & Bethge (2019) |
| BagNet33 Brendel & Bethge (2019) |
| ResNet50 trained on SIN Geirhos et al. (2022) |
| ResNet50 trained on SIN and IN Geirhos et al. (2022) |
| ResNet50 trained on SIN and IN, finetuned on IN Geirhos et al. (2022) |
| SimCLR ResNet50 x1 T. Chen et al. (2020) |
| SimCLR ResNet50 x2 T. Chen et al. (2020) |
| SimCLR ResNet50 x4 T. Chen et al. (2020) |

Table 6: **List of all 169 3D Object Models utilized in constructing benchmark.**

| Model Name | Model Class | License | URL |
|---|---|---|---|
| Airplane 1 | Airplane | CC BY | Source URL |
| Airplane 10 | Airplane | CC BY | Source URL |
| Airplane 11 | Airplane | CC BY | Source URL |
| Airplane 2 | Airplane | CC BY | Source URL |
| Airplane 3 | Airplane | CC BY | Source URL |
| Airplane 4 | Airplane | CC BY | Source URL |
| Airplane 5 | Airplane | CC BY-NC-SA | Source URL |
| Airplane 6 | Airplane | CC BY | Source URL |
| Airplane 7 | Airplane | Sketchfab Standard | Source URL |
| Airplane 8 | Airplane | CC BY | Source URL |
| Airplane 9 | Airplane | CC BY-NC | Source URL |
| Bear 1 | Bear | CC BY | Source URL |
| Bear 10 | Bear | TurboSquid | Source URL |
| Bear 11 | Bear | CC BY | Source URL |
| Bear 2 | Bear | TurboSquid | Source URL |
| Bear 3 | Bear | TurboSquid | Source URL |
| Bear 4 | Bear | Sketchfab Standard | Source URL |
| Bear 5 | Bear | CC BY | Source URL |
| Bear 6 | Bear | CC BY | Source URL |
| Bear 7 | Bear | Sketchfab Standard | Source URL |
| Bear 8 | Bear | TurboSquid | Source URL |
| Bear 9 | Bear | TurboSquid | Source URL |
| Bicycle 1 | Bicycle | Sketchfab Standard | Source URL |
| Bicycle 10 | Bicycle | CC BY | Source URL |
| Bicycle 11 | Bicycle | CC BY | Source URL |
| Bicycle 2 | Bicycle | CC BY-SA | Source URL |
| Bicycle 3 | Bicycle | BlenderKit Royalty Free | Source URL |
| Bicycle 4 | Bicycle | CC BY | Source URL |
| Bicycle 5 | Bicycle | CC BY | Source URL |
| Bicycle 6 | Bicycle | CC BY | Source URL |
| Bicycle 7 | Bicycle | CC BY | Source URL |
| Bicycle 8 | Bicycle | CC BY | Source URL |
| Bicycle 9 | Bicycle | CC BY-NC-SA | Source URL |
| Bird 1 | Bird | CC BY-NC | Source URL |
| Bird 10 | Bird | Sketchfab Standard | Source URL |
| Bird 11 | Bird | TurboSquid | Source URL |
| Bird 2 | Bird | CC BY | Source URL |
| Bird 3 | Bird | CC BY | Source URL |
| Bird 4 | Bird | CC BY-NC | Source URL |
| Bird 5 | Bird | CC BY | Source URL |
| Bird 6 | Bird | TurboSquid | Source URL |
| Bird 7 | Bird | TurboSquid | Source URL |
| Bird 8 | Bird | CC BY | Source URL |
| Bird 9 | Bird | TurboSquid | Source URL |
| Boat 1 | Boat | CC BY-NC-ND | Source URL |
| Boat 10 | Boat | Sketchfab Editorial | Source URL |
| Boat 11 | Boat | Sketchfab Editorial | Source URL |
| Boat 2 | Boat | Sketchfab Editorial | Source URL |
| Boat 3 | Boat | TurboSquid | Source URL |
| Boat 4 | Boat | Sketchfab Standard | Source URL |

Table 7: **List of all 169 3D Object Models utilized in constructing benchmark.**

| Model Name | Model Class | License | URL |
|---|---|---|---|
| Boat 5 | Boat | Sketchfab Standard | Source URL |
| Boat 6 | Boat | CC BY | Source URL |
| Boat 7 | Boat | CC BY | Source URL |
| Boat 8 | Boat | Sketchfab Standard | Source URL |
| Boat 9 | Boat | Sketchfab Editorial | Source URL |
| Bottle 1 | Bottle | CC0 | Source URL |
| Bottle 10 | Bottle | BlenderKit Royalty Free | Source URL |
| Bottle 11 | Bottle | CC BY | Source URL |
| Bottle 2 | Bottle | CC BY | Source URL |
| Bottle 3 | Bottle | Sketchfab Standard | Source URL |
| Bottle 4 | Bottle | CC BY | Source URL |
| Bottle 5 | Bottle | BlenderKit Royalty Free | Source URL |
| Bottle 6 | Bottle | BlenderKit Royalty Free | Source URL |
| Bottle 7 | Bottle | Sketchfab Standard | Source URL |
| Bottle 8 | Bottle | BlenderKit Royalty Free | Source URL |
| Bottle 9 | Bottle | CC BY | Source URL |
| Car 1 | Car | CC BY | Source URL |
| Car 10 | Car | CC BY | Source URL |
| Car 11 | Car | CC BY | Source URL |
| Car 2 | Car | CC BY-NC-ND | Source URL |
| Car 3 | Car | CC BY | Source URL |
| Car 4 | Car | CC BY | Source URL |
| Car 5 | Car | CC BY-NC | Source URL |
| Car 6 | Car | CC BY-NC | Source URL |
| Car 7 | Car | CC BY | Source URL |
| Car 8 | Car | CC BY | Source URL |
| Car 9 | Car | CC BY | Source URL |
| Cat 1 | Cat | CC BY | Source URL |
| Cat 10 | Cat | TurboSquid | Source URL |
| Cat 11 | Cat | CC BY | Source URL |
| Cat 2 | Cat | CC BY | Source URL |
| Cat 3 | Cat | CC BY | Source URL |
| Cat 4 | Cat | TurboSquid | Source URL |
| Cat 5 | Cat | CC BY | Source URL |
| Cat 6 | Cat | Sketchfab Standard | Source URL |
| Cat 7 | Cat | TurboSquid | Source URL |
| Cat 8 | Cat | TurboSquid | Source URL |
| Cat 9 | Cat | CC BY | Source URL |
| Chair 1 | Chair | CC BY | Source URL |
| Chair 10 | Chair | CC0 | Source URL |
| Chair 11 | Chair | CC BY | Source URL |
| Chair 2 | Chair | CC BY | Source URL |
| Chair 3 | Chair | CC BY | Source URL |
| Chair 4 | Chair | CC BY | Source URL |
| Chair 5 | Chair | CC BY | Source URL |
| Chair 6 | Chair | CC BY | Source URL |
| Chair 7 | Chair | CC BY | Source URL |
| Chair 8 | Chair | CC BY | Source URL |
| Chair 9 | Chair | CC BY | Source URL |
| Clock 1 | Clock | CC0 | Source URL |
| Clock 10 | Clock | BlenderKit Royalty Free | Source URL |
| Clock 11 | Clock | BlenderKit Royalty Free | Source URL |
| Clock 2 | Clock | CC0 | Source URL |
| Clock 3 | Clock | BlenderKit Royalty Free | Source URL |

Table 8: **List of all 169 3D Object Models utilized in constructing benchmark.**

| Model Name | Model Class | License | URL |
|---|---|---|---|
| Clock 7 | Clock | BlenderKit Royalty Free | Source URL |
| Clock 8 | Clock | CC0 | Source URL |
| Clock 9 | Clock | BlenderKit Royalty Free | Source URL |
| Clock 4 | Clock | BlenderKit Royalty Free | Source URL |
| Clock 5 | Clock | BlenderKit Royalty Free | Source URL |
| Clock 6 | Clock | CC BY | Source URL |
| Dog 1 | Dog | CC BY | Source URL |
| Dog 10 | Dog | Sketchfab Standard | Source URL |
| Dog 11 | Dog | CC BY | Source URL |
| Dog 2 | Dog | CC BY | Source URL |
| Dog 3 | Dog | Sketchfab Standard | Source URL |
| Dog 4 | Dog | TurboSquid | Source URL |
| Dog 5 | Dog | CC BY | Source URL |
| Dog 6 | Dog | TurboSquid | Source URL |
| Dog 7 | Dog | TurboSquid | Source URL |
| Dog 8 | Dog | TurboSquid | Source URL |
| Dog 9 | Dog | TurboSquid | Source URL |
| Elephant 1 | Elephant | Sketchfab Standard | Source URL |
| Elephant 10 | Elephant | CC BY | Source URL |
| Elephant 11 | Elephant | Sketchfab Editorial | Source URL |
| Elephant 2 | Elephant | Sketchfab Standard | Source URL |
| Elephant 3 | Elephant | CC BY | Source URL |
| Elephant 4 | Elephant | CC BY | Source URL |
| Elephant 5 | Elephant | CGTrader Royalty Free | Source URL |
| Elephant 6 | Elephant | Sketchfab Standard | Source URL |
| Elephant 7 | Elephant | Sketchfab Standard | Source URL |
| Elephant 8 | Elephant | CC BY | Source URL |
| Elephant 9 | Elephant | Sketchfab Standard | Source URL |
| Keyboard 1 | Keyboard | CC BY | Source URL |
| Keyboard 10 | Keyboard | CC BY | Source URL |
| Keyboard 11 | Keyboard | CC BY | Source URL |
| Keyboard 2 | Keyboard | BlenderKit Royalty Free | Source URL |
| Keyboard 3 | Keyboard | CC BY | Source URL |
| Keyboard 4 | Keyboard | CC BY | Source URL |
| Keyboard 5 | Keyboard | BlenderKit Royalty Free | Source URL |
| Keyboard 6 | Keyboard | BlenderKit Royalty Free | Source URL |
| Keyboard 7 | Keyboard | CC BY | Source URL |
| Keyboard 8 | Keyboard | CC BY | Source URL |
| Keyboard 9 | Keyboard | BlenderKit Royalty Free | Source URL |
| Knife 1 | Knife | CC BY | Source URL |
| Knife 10 | Knife | CC BY | Source URL |
| Knife 11 | Knife | CC BY | Source URL |
| Knife 2 | Knife | CC BY | Source URL |
| Knife 3 | Knife | Sketchfab Standard | Source URL |
| Knife 4 | Knife | Sketchfab Standard | Source URL |
| Knife 5 | Knife | CC BY | Source URL |
| Knife 6 | Knife | CC BY | Source URL |
| Knife 7 | Knife | CC BY | Source URL |
| Knife 8 | Knife | CC BY | Source URL |
| Knife 9 | Knife | CC BY | Source URL |
| Oven 1 | Oven | TurboSquid | Source URL |
| Oven 10 | Oven | TurboSquid | Source URL |
| Oven 11 | Oven | TurboSquid | Source URL |

Table 9: **List of all 169 3D Object Models utilized in constructing benchmark.**

| Model Name | Model Class | License | URL |
|---|---|---|---|
| Oven 2 | Oven | TurboSquid | Source URL |
| Oven 3 | Oven | TurboSquid | Source URL |
| Oven 4 | Oven | TurboSquid | Source URL |
| Oven 5 | Oven | TurboSquid | Source URL |
| Oven 6 | Oven | TurboSquid | Source URL |
| Oven 7 | Oven | TurboSquid | Source URL |
| Oven 8 | Oven | TurboSquid | Source URL |
| Oven 9 | Oven | TurboSquid | Source URL |
| Truck 1 | Truck | Sketchfab Editorial | Source URL |
| Truck 10 | Truck | CC BY | Source URL |
| Truck 11 | Truck | CC BY | Source URL |
| Truck 2 | Truck | Sketchfab Editorial | Source URL |
| Truck 3 | Truck | CC BY-ND | Source URL |
| Truck 4 | Truck | Sketchfab Editorial | Source URL |
| Truck 5 | Truck | TurboSquid | Source URL |
| Truck 6 | Truck | Sketchfab Standard | Source URL |
| Truck 7 | Truck | CC BY | Source URL |
| Truck 8 | Truck | CC BY | Source URL |
| Truck 9 | Truck | CC BY | Source URL |

