# OpenReview forum: "Robustness to 3D Object Transformations in Humans and Image-Based Deep Neural Networks"
_ccneuro.org/CCN/2025/Proceedings — CCN 2025 Proceedings asProceedingsTalkPoster_

### Official Review · Reviewer_52kF · 2025-03-31
**Review of "Robustness to 3D Object Transformations in Humans and Deep Neural Networks"**

**Soundness:** 3
**Clarity:** 2

**Comments:**

This paper examines how both humans and deep neural networks categorize objects across 3D rotations, a core perceptual capacity. The paper is of central interest to the CCN community in that it seeks to understand how object representations and processing differ between humans and state-of-the-art ANNs.  It represents an advance over previous work by both (1) providing a novel dataset for benchmarking future models, and (2) evaluating a large suite of computational models against this benchmark. I think that the paper is of high interest to the scientific community at CCN, and ties into broad, relevant debates (Conwell et al., 2024) about the amount of data and kinds of architectures needed to produce human-like object representations.  Strengths of this paper include (1) a well thought-out human behavioral experiment, (2) the use of a large suite of modern computational models, and a good description of the differences among them, (3) clear writing and headings that guide the reader through the paper, as well as nice diagrams and visualizations of the experimental manipulations. Major concerns follow:

However, I’m not sure that I’m convinced of one of the main claims of the paper, stated in line 488: “The fact that data scale, rather than architecture or vision-language training, is the primary driver of model improvements suggests that increasing dataset size can bridge some gaps, but not all.” This claim seems to rest on the fact that while VLMs show the highest performance, the more complex CLIP models with larger training datasets (Figure 5) are the best predictors of 16-way categorization.  But of course this doesn’t necessarily suggest that it is only dataset size – independent of vision-language training — that is going to lead to improved robustness to 3D transformations. Instead, it seems that some vision models trained with larger dataset sizes (Figure 3a, b) can rival the performance of the CLIP models. Perhaps this was done systematically, and there is a way that we can directly compare the size of the training regimes in the vision-only vs. CLIP models, but as it stands it feels more like a conjecture (and possibility) rather than a take-home conclusion—and as a side note, it’s pretty hard to find the result in Figure 3a,b to piece it the argument together. This is my main issue with the soundness of the paper—if the authors want to contribute to this debate, then I think that this evidence and analyses needs to be elaborated upon.

In the same vein, for Figure 5: It would be useful to plot dataset size as a continuous measure against accuracy, in order to evaluate what the saturation point is. In Figure 5, some of the aspects of the plot are undefined – e.g. the difference between the light blue vs dark blue portions (I think canonical vs. transformed views, inferring from Figure 3?). Also, I understand that the red dashed line is human performance, but what is the pink dashed line? If this refers to some sort of noise ceiling, that should be indicated so readers can understand.

This issue comes back up again in Figure 6, where it seems that the comparison between models and humans is only made at a qualitative level by examining trends in item effects. Here – if the authors want to stipulate that a vision++ trained model is as good as a VLM – it would be nice to see those particular models (vs. a VLM CLIP model, trained to a lesser degree?) pitted against each other. Instead, we see averages across the broad classes of models, which feel relatively unhelpful except for highlighting that (1) there is variability and (2) VLM’s are doing quite well. I understand that some of the key comparisons for these models are in the model zoo in the Appendix, but I think they should be brought to the main text.

And for Figure 4: The models are compressed into one average line, which is general helpful for seeing trends—but we’ve lost information here about the variabilities in models within each class. Could individual model performances be reported using a low alpha individual lines, perhaps in an Appendix plot? I’m curious whether the paper’s classification of vision / vision + / vision ++ aligns directly with performance in all of the different domains. This is another domain where drawing out a specific comparison between the models that make the key point about dataset size vs. VLM training would be helpful.

On another note, I’m also a little uncomfortable with the Fellbaum 2010 categories as a proxy for all of object recognition capacities. In my mind, these feel relatively limited. I understand that 16AFC affords quite an advantage, but some of these categories (e.g., “dog,”bear”) are much more common than others (“keyboard”). Some of the stimuli have quite a bit of perceptual variability (there are diverse clock exemplars, and chair exemplars) whereas other classes are much more homogenous (e.g., the bears). I’m not asking that the authors redo the experiments, but I think that a statement on the limitations of generalizability towards the end of the paper is warranted. More broadly, the General Discussion is pretty short and I think could be expanded to include more nuance.

Overall, I was excited about this paper, but I feel it needs a bit of restructuring and additional analyses to make the main points clear.

Minor concerns:
1. This phrase - “We speculate that the models’ visual diet may contain more images of objects rotated about their vertical than their horizontal axis.” — feels like it perhaps belongs in a General Discussion whether the authors could talk about directions for future work, as this is something that could be investigated directly.
2. There is quite a bit of text devoted to the exact methods of stimuli generation, which I appreciate—but I could have used more details on how the models were evaluate. For example,  why were these ~30 models selected, and why were the other many other models culled from this analyses?
3. The institution is listed in Human Data Collection (Line 212). This is breach of anonymity.

**Expertise:**

3

**Interest:**

3

---

> ### Author Rebuttal · Authors · 2025-04-15
>
> Thank you for your thoughtful feedback and for recognizing the potential of our study. We appreciate your positive remarks regarding our novel dataset, and our extensive evaluations.
>
> # Clarifying the influence of data scale versus vision-language training
> We agree that stronger conclusions about the relative importance of dataset size, supervision type, and architecture would require more controlled comparisons. Due to limited model availability, we could not fully disentangle these dimensions across the full model set.
>
> In response, we have softened our wording in the manuscript to better reflect the evidence. Notably, Figure 5 provides a controlled comparison across CLIP models with matched architecture and training objective, showing that increasing dataset size leads to clear gains in accuracy. Additionally, Figure 3 shows that a vision-only++ model trained with DINOv2 on 142M images performs comparably to some vision-language models (and drastically better than DINO with 1.28M), suggesting that data richness—independent of language supervision—plays a substantial role in robustness.
>
> # Adding more detail on individual model variability
> We have followed your advice by including individual model performance overlays in supplementary figures (Figure 12). Although the aggregated performance lines in Figure 4 still help highlight cross-category trends, the supplementary plots offer a clear view of the variability within each model class. We have also included a supplementary figure associated with Figure 6, which shows category-level robustness alignment (Pearson r) between each model and the human participants (Figure 14).
>
> # Figure 5 representation of dataset size and other references
> Thank you for pointing out the ambiguity in our original figure design. We have updated Figure 5 to plot dataset size and number of parameters as a continuous variable against accuracy. We also clarify the distinction between canonical and transformed views more clearly, and refine our color-coding to improve readability.
>
> # Limited scope of 16 categories
> We acknowledge that the 16 categories do not capture the full complexity of real-world object recognition. In our revised limitations section, we explicitly note this constraint and encourage caution in extrapolating from these results to more fine-grained categories. We also highlight the need for future work to incorporate broader and more diverse categories to capture the full variability of human object perception.

---

> > ### Comment · Reviewer_52kF · 2025-04-22
> >
> > Thanks for engaging with my comments.  I have also been following the comments of the other reviewer's, and do think that stating the limitations of the current findings to image-based models is useful. However, I also still recognize that many many models the CCN community uses are image-based--in particular those used to model responses in the ventral stream-- and so understanding how well they compare to humans on 3D object recognition benchmarks will be important to move the field forward.
> >
> > With respect to my own review, I appreciate the revisions to the writing which tones down the implications of their findings in several spots throughout the paper. However, on this read I noticed the term "data richness", and wanted to leave a small note—as "data-richness" and "training dataset size" are not necessarily equivalent  (data richness is the phrase used in the abstract/GD)— in principle, they can be disassociated, and have different implications. In my mind, a "rich" dataset might be taken to one that shows category exemplars across a broad diversity of contexts with many exemplars (and perhaps with some thoughtfulness with respect to ecological validity). In contrast, one could have a large dataset that focused exclusively on a certain kind of categorization task, e.g., dog breeds. However, data richness/dataset size are probably confounded in many training datasets.  My practical ask is that the author's think through on these terms as they present/write about this work in the future.
> >
> > In addition, I think that in the future it will be useful for the author's not only to explore which categories show variation across humans/models, but also why—what kinds of experience do models/humans have, and how different are they? Do we experience rotations of some categories more than others--for example, small, manipulable objects vs. large, stationary objects (e.g., couches)? I don't need this to be incorporated into a revision, but I would encourage the author's (as per another reviewer's comment) to think more about what these kinds of findings can tell us about both human vision and how these capacities are built in the first place. I appreciated the new section in the discussion on "implications for human vision" in response to the other reviewer.
> >
> > I have updated my scores such that the "soundness" rating is a point higher. I look forward to seeing this work at CCN!

---

> > > ### Author Response · Authors · 2025-04-22
> > >
> > > We thank the reviewer for their thoughtful comments and suggestions. We agree that data richness and training dataset size are not necessarily equivalent and will take this into account in future presentations of this work. We also appreciate the reviewer's encouragement to explore why humans and models differ on category errors and confusions, and what our findings tell us about human vision / human visual development.

---

### Official Review · Reviewer_mLq8 · 2025-04-01
**Well-conducted study taking stock of human-likeness of 3D object recognition in current vision and vision language models**

**Soundness:** 2
**Clarity:** 3

**Comments:**

This is a well-conducted study taking stock of the 3D object recognition abilities of current vision and vision-language models, comparing their accuracy to human behavior.

The authors test a collection of vision and vision-language models to recognize 3D objects under various systematic variations, including spatial shifts, scale, pitch, yaw, and backgrounds. Human behavior is tested in a masked 16AFC design. The authors demonstrate that large data-trained vision and vision-language models approach human object recognition capabilities, both in terms of overall accuracy and their robustness to variations in irrelevant dimensions.

The work is of interest to cognitive scientists studying human visual recognition and may also be of interest to machine learning researchers seeking to align models with human perception. The experimental methods are sound (see the comments below for more details). The large dataset, comprising 220 participants each with 1.5 hours of behavioral data, is commendable and could be a valuable resource for the community if the authors decide to share it. The paper is very well written.

Comments:

(1) The study can be viewed as a benchmark for models and an assessment of a specific aspect of human-model alignment in vision. While some implications of the results for models were discussed (e.g., more data leads to better alignment), I was missing a reflection on what this all means for human vision. Humans and VLMs exhibit very similar behavior. Does that mean human vision can be explained as being trained with big data (via evolution and experience), providing a large bank of templates for fast recognition of 3D objects in all kinds of poses? The parallelism of VLM and humans suggests just that – there is not much more to human 3D object recognition than a very well-trained bank of filters for all poses. How does that contrast with existing theories of 3D object recognition? For a broader impact, I hope the authors will go deeper in subsequent work.

(2) A slight limitation of the experiments is that they appear to be too easy for humans – accuracy is consistently very high. There is only little room for probing error consistency between humans and the best-performing models. In more challenging conditions, differences between models and humans may become apparent, which are currently obscured.

(3) Humans experience the same objects repeatedly in the experiment (169 objects over the course of more than 1280 trials), whereas models are memory-less. Is the small gap between humans and VLMs left explainable by the memory benefit of humans? Could you provide evidence for this (e.g., by examining only early trials)?

(4) Will the behavioral data also be released with the model evaluations? I was unable to find a statement on this matter.

**Expertise:**

3

**Interest:**

2

---

> ### Author Rebuttal · Authors · 2025-04-15
>
> Thank you very much for the thorough and constructive review of our work. We appreciate your comments about discussing the implications of the results for human vision.
> # Implications for human vision
>
> You raise an excellent point regarding the implications of our findings for theories of human object recognition. Our results indeed highlight similarities between VLMs and human behavior, suggesting that extensive visual exposure—akin to training on large datasets—may play a crucial role in shaping human visual recognition capabilities. In the revised paper, we further discuss how our findings align with a view-based account of recognition (e.g., Tarr and Poggio).
>
> To further explore these representational strategies, we also conducted additional analyses of category confusion patterns. As depicted in Figure 14, we analyzed correlations of off-diagonal entries in confusion matrices across human participants and model categories for each transformation. Across transformations, human participants consistently showed greater agreement with one another in error patterns than with any model, while models correlated more strongly amongst themselves (Figures 14 & 15). This analysis indicates that despite comparable accuracy, models may employ representational strategies different from humans.
>
> # Potential human memory effects
>
> We appreciate the concern that human participants might benefit from repeated exposures to the same objects. We examined performance across consecutive experimental blocks (see newly added Figure 13), each comprising 128 trials, allowing us to compare accuracy in early vs. later blocks. While we observed marginal performance gains for human participants on a few rotations (e.g., X and Z), overall accuracy remained relatively stable across all blocks. This suggests that memory-driven improvement over time was minimal. Consequently, the modest performance gap between humans and VLMs in our study does not appear to stem primarily from repeated exposures or memory effects in humans.
>
> # Data availability
>
> We are committed to releasing both the human behavioral data and model evaluations during the camera-ready period. A link to the dataset will provided on line 197: “are available in the supplementary codebase: GitHub.com/x.”. The codebase will include:
> - Anonymized complete human trial-level data for all 220 participants
> - Stimuli used in the experiment
> - Model predictions for all 169 models across all transformations and objects

---

> > ### Comment · Reviewer_mLq8 · 2025-04-18
> >
> > Thank you for your thoughtful responses and the additional analyses.
> >
> > I have just one remaining request. The legends for Figures 14 - 16 are a bit confusing and potentially contain errors (?). Please expand and clarify them, and explicitly indicate how the three figures relate—for example, do the bars in Figure 14 correspond to the diagonal entries in Figure 15 (?). What is the difference between 14 and 16? Also in 14: "While VLMs showed comparable performance on our invariant object recognition benchmark, VLMs correlations remain moderate, indicating persistent differences in the specific errors made by humans versus VLMs". The VLM correlations are higher than those for humans in most plots and Figure 14 doesn't really speak to whether there are persistent differences in the errors made by humans vs VLMs as there is no VLM-human correlation displayed.

---

> > > ### Author Response · Authors · 2025-04-20
> > >
> > > We thank the reviewer for their feedback. We apologize - we accidentally used the same legend for Figs. 14 and 16. We have expanded and clarified the figure legends as follows:
> > >
> > > "Figure 14. Model-to-human alignment on category difficulty for each identity-preserving transformation. We computed robustness scores across object categories (16-element vectors) and correlated these scores between individual models and individual humans. Bars show the average model-to-human correlations (Pearson r) for individual models. The red dashed line indicates the average human-to-human correlation, with shaded areas indicating 95% confidence intervals. Stronger positive correlations indicate stronger agreement on which categories are challenging.
> > >
> > > Figure 15. Alignment on category confusions, within and between humans and models, for each identity-preserving transformation. We computed category confusion matrices for models and humans, whose off-diagonal entries indicate how often each true category is confused with other categories. We correlated the off-diagonal entries of these matrices between individual models and individual humans, and summarized the results - averaging across humans and across models within a family - in the displayed correlation matrices. Stronger positive correlations indicate stronger agreement on category confusions.
> > >
> > > Figure 16. Model-to-human alignment on category confusions for each identity-preserving transformation. We computed category confusion matrices for models and humans, whose off-diagonal entries indicate how often each true category is confused with other categories. We correlated the off-diagonal entries of these matrices between individual models and individual humans. We here show alignment between humans and individual models (first row/column of the matrices shown in  Figure 15, but not averaged across models within a family). Bars show the average model-to-human correlations (Pearson r) for individual models. The red dashed line indicates the average human-to-human correlation, with shaded areas indicating 95% confidence intervals. Stronger positive correlations indicate stronger agreement on category confusions."
> > >
> > > Fig. 14 complements Fig. 5 -- both are based on category-specific robustness scores, but whereas Fig. 5 provides a visual overview (averaging those scores within humans and within models in a family), Fig. 14 provides a quantitative analysis at the level of individual models focusing on model-to-human alignment (showing correlations of scores between individual models and humans).
> > >
> > > Figs. 5 and 14 evaluate whether models and humans align on which categories are challenging, while Figs. 15 and 16 evaluate whether models and humans align on category confusions (when errors are made, do they confuse the same categories?).

---

### Official Review · Reviewer_hgMz · 2025-04-01
**Clearly-written, solid work, but issues with model selection limit relevance**

**Soundness:** 2
**Clarity:** 3

**Comments:**

This paper summarizes results from a comparison of humans and various (almost exclusively) image-based vision (or vision+language) models on the categorization of parametrically generated 3D objects, shown to both (time-limited) humans and machines at various rotations (in plane and depth) and different contexts / background. The primary intent of the paper appears to be a more systematic comparison of robust / context-independent / rotation-invariant category recognition -- and the authors do indeed deliver on this! The analysis is straightforward and well-motivated, and the write-up is crystal clear. Thus, as pertains strictly to analytic / empirical methodology ("soundness") and presentation ("clarity"), I'm happy to say "no notes!"

In terms of relevance / "interest", there are what I consider to be a few substantial issues, almost all of which pertain to model selection. In short, effectively all of the models chosen by the authors are (as far as I can tell) models trained exclusively on static, naturalistic images. There are no video models, 3D-aware or point-cloud multimodal models, monocular depth models, 3D segmentation models, NERF models, generative shape models, et cetera... despite many such pretrained models existing and often readily available for download (e.g. ULIP, MV-MAE, ShapeNet, MiDaS, MV / GV CNN, VLN-Bert, BevFormer, Embodied Transformer, GQN, Mask3D, HAMT). To be fair, this is most certainly not an issue that undermines the work in terms of "soundness" (see above), but it does dramatically limit the work's relevance. Without these models, the work feels effectively like a more comprehensive, direct, and well-designed recapitulation of known phenomena: large vision models / vision-language foundation models trained on more data are more robust human-like in their recognition behavior, training data (empirically isolated) matters more than architecture to "close" the gap on human behavior, and models still show greater "difficulty" recognizing objects in "out-of-distribution" views than humans do (despite improvements with increasing training dataset size). The authors mention in the last paragraph of their discussion that more focus is needed on developing models "more robust to identity-preserving transformations", but the machine learning community has rather actively been pursuing precisely this kind of research for awhile now, and has been making nontrivial, even rather substantive gains. Limiting the assessment to static image models thus somewhat dramatically disregards the possibility that the gap between humans and models in terms of invariant recognition may well have been closed by something other than internet-scale training.

All in all, I would still say the clarity, comprehension, and quality of the analyses in this work mean it should still be considered for acceptance, but I do actively encourage the authors to consider assessing other kinds of models during the rebuttal period, or at least to dramatically expand the discussion to note the limitation of scope inherent to the model selection.

**Expertise:**

2

**Interest:**

2

---

> ### Author Rebuttal · Authors · 2025-04-15
>
> We thank the review for their thorough review and valuable feedback on our submission. We appreciate the time and effort invested in evaluating our work.
>
> # Specialized Models Evaluation
>
> We thank the reviewer for pointing out existing specialized models (e.g. ULIP, MV-MAE, ShapeNet, MiDaS, etc…) that we could evaluate. We agree that evaluating these specialized models on invariant object recognition would be a valuable contribution. However, the primary challenge with incorporating these specialized models is that they cannot be directly evaluated on our 16-way categorization task without substantial modifications. Many of these models have specialized architectures designed for specific tasks (e.g., depth estimation, 3D classification) rather than object recognition, and would require additional components like fine-tuning to produce the category labels needed for our evaluation framework. Furthermore, most of these models require different inputs (e.g. videos, 3D models, etc…), making them hard to fairly compare with vision models with static images.
>
> These modifications would introduce confounding variables that could make fair comparisons difficult, as performance would depend not only on the models' inherent representational capabilities but also on the quality of the adaptation method.
>
> We would like to also clarify that among the 169 models we investigated (Tables 2-5), we did include various specialized architectures and training regimes, including ShapeNet models trained with Stylized-ImageNet, BagNets (designed for local feature processing), weakly-supervised ResNets, adversarially robust ResNets, and Masked MAE variants. However, during our analysis, we found that these specialized models did not come close to the performance of the standard vision and vision-language models reported in our paper when evaluated on our benchmark (Figures 17-19).
>
> Based on the reviewer's feedback, we have added a comprehensive limitations section to the paper that addresses the scope of our model selection. This section explicitly acknowledges that our study focused primarily on static images, and discusses the challenges of incorporating specialized 3D-aware models into our evaluation framework. Furthermore, to better illustrate the breadth of models we explored, we have included appendix figures that show all 169 evaluated models' accuracy, robustness, and error consistency (Figures 17-19).

---

> > ### Comment · Reviewer_hgMz · 2025-04-17
> >
> > I thank the authors' for their response, and appreciate the updates to the paper acknowledging limitations.
> >
> > However, my concern regarding the limited scope of model / architecture choice is now amplified with the authors' noting that their benchmark cannot be applied to other architectures. If I've understood correctly, the primary "test" in the benchmark is 16-way categorization. The authors are correct that models not otherwise designed for classification would require additional components, but at base, this additional component is effectively a single linear classifier applied to the feature vectors that are extractable from almost any model -- including many of the models I mention in my primary review. (Video models are indeed a more difficult case, but setting those aside for now, there do seem to be many other kinds of models that could be evaluated on the basis of the single images that comprise the 16-way categorization task).
> >
> > I understand time limitations may not allow the authors to update their work to address this limitation now with new analyses, but I would ask that even further specification of limitation be made (perhaps in the paper's title, even). The key limitation that needs underscoring, basically, is that this evaluation has been applied uniquely to "image-recognizing deep neural networks" (or something like this).

---

> > > ### Author Response · Authors · 2025-04-21
> > >
> > > We thank the reviewer for their thoughtful second-round comments and for engaging deeply with the scope and model selection aspects of our work.
> > >
> > > We agree that a key limitation of our study is the focus on deep neural networks trained on single static images, and that many models trained on multiview, temporal, or 3D data could in principle be evaluated on our benchmark via linear probing. We also agree that this point should be made even more explicit in the manuscript, and we have made several changes to do so.
> > >
> > > We have revised the title to:
> > > "Robustness to 3D Object Transformations in Humans and Image-Based Deep Neural Networks"
> > > We chose “image-based” to reflect that our benchmark evaluates models that process single 2D images as input, rather than those trained on video, multiview, or 3D data. After careful consideration, we opted not to use the term “image-recognizing,” as it may unintentionally suggest exclusion of models trained with self-supervised objectives that learn visual representations without explicit categorization during pretraining. This includes both vision-language models (e.g., CLIP) and vision-only models trained with contrastive or masked autoencoding objectives. These models do not explicitly perform classification during training but are evaluated on our benchmark using readily available linear probes or zero-shot classification.
> > >
> > > In the abstract, we now explicitly state the scope of included models:
> > > “Here, we evaluate both humans and image-based deep neural networks, including vision-only and vision-language models trained with supervised, self-supervised, or weakly supervised objectives…”
> > > We also note in the final lines of the abstract that:
> > > “Our benchmark excludes models trained on video, multiview, or 3D data, but is in principle compatible with such models and may support their evaluation in future work.”
> > >
> > > In the introduction, we have added a clarifying sentence:
> > > “We focus on deep neural networks that take a single static image as input and support object categorization, including vision-only and vision-language models trained with (weakly) supervised or self-supervised objectives.”
> > >
> > > To directly address the reviewer's suggestion, we have added more detailed and clearly located acknowledgments of the scope of our model evaluation in three places in the discussion:
> > >
> > > [Implications for modeling human vision] “While we focused on models trained to process single static images, recent advances in video models, point-cloud methods, and generative shape representations offer promising new directions. Although these models were not evaluated here, several could, in principle, be adapted to our benchmark using extracted features and linear probing. Their inclusion will be important for testing whether remaining human-model differences reflect architectural limitations, training exposure, or representational format.”
> > >
> > > [Limitations]: “Our model set spans a wide range of architectures and training regimes, but is limited to deep neural networks that process single static images and support object categorization. This excludes models trained on multiview, temporal, or 3D input, such as video transformers, NeRF-based systems, and point-cloud models. While many of these models were developed for tasks other than categorization, several could be adapted for evaluation on our benchmark using linear probing. Their inclusion in future work will be important for assessing the generality of our findings.”
> > >
> > > [Conclusion]: “While our findings shed light on the capabilities of models trained on single static images, we did not evaluate models explicitly designed for 3D object understanding. Incorporating such models will be essential for testing whether richer structural priors and training signals can further narrow the gap. With continued progress and broader model evaluation, models may come closer to capturing the core computational principles that support robust human object recognition across the complexity of the real world.”
> > >
> > > Together, we hope these clarifications clearly acknowledge the scope of our model comparisons and highlight opportunities for future benchmarking of broader model classes.

---

### Meta-Review · Area_Chair_TBUC · 2025-05-03

**Ccn Recommendation:** Accept as Proceedings

**Metareview:**

During rebuttal and discussions, the authors acknowledged these limitations and have revised the paper by clearly explaining these limitations.
Regarding the second limitation mentioned in the Summary, the authors noted they could not fully disentangle these dimensions across the complete model set due to limited model availability. However, they mentioned that results presented in Figures 3 and 5 partially support the importance of dataset size compared to bimodal training data in CLIP (a VLM) and DINO (a vision-only model).

After reviewing the comments and discussions, and acknowledging the potential limitations of the work, I believe this study still provides a valuable contribution to the field, given the widespread use of image-trained models by the CCN community as models of the human visual system. Therefore, I recommend **acceptance** for this paper.

**Summary:**

This paper compares the robustness of human vision and artificial neural networks against 3D object transformations (translation, scaling, and rotation). All reviewers agreed on the clarity of the writing and the soundness of the methodology. While all reviewers were supportive of the paper and affirmed its important contributions to cognitive neuroscience and potentially AI communities, they raised concerns regarding the scope and claims that could be made with the current analyses and dataset.

Two major concerns emerged:

1. The study includes only ANN models trained in supervised or self-supervised fashion on images, while excluding recent important developments in machine vision. These include models trained on videos (providing more exposure to 3D information) and 3D-aware model architectures explicitly designed to work with multiple views. As one reviewer noted, the gap between humans and AI models may have already been closed by innovations beyond mere training dataset size. These comparisons are not included in this work.

2. The claim that VLMs are most aligned with human vision in terms of 3D transformation robustness could be attributed either to VLMs' substantially larger training datasets compared to other models, or to the use of language in training. In its current form, the paper cannot distinguish between these two possibilities.

**Expertise:**

3